# Water depth estimate and flood extent enhancement for satellite-based inundation maps

Andrea Betterle[1] and Peter Salamon[1]

[1]European Commission, Joint Research Centre, Ispra (Italy)

**Correspondence:** Andrea Betterle (andrea.betterle@ec.europa.eu)

**Abstract.**

Floods are extreme hydrological events that can reshape the landscape, transform entire ecosystems, and alter the relationship between living organisms and the surrounding environment. Every year, fluvial and coastal floods claim thousands of human lives, cause enormous direct damages and inestimable indirect losses, particularly in less developed and more vulnerable regions. Monitoring the spatio-temporal evolution of floods is fundamental to reduce their devastating consequences. Observing floods from space can make the difference: from this distant vantage point it is possible to monitor vast areas consistently and, by leveraging multiple sensors on different satellites, it is possible to acquire a comprehensive overview on the evolution of floods at a large scale. Synthetic Aperture Radar sensors (SAR), in particular, have proven extremely effective for flood monitoring, as they can operate day and night and in all weather conditions, with a highly discriminatory power. On the other hand, SAR sensors are unable to reliably detect water in some cases, the most critical being urban areas. Furthermore, flood water depth – which is a fundamental variable for emergency response and impact calculations – cannot be estimated remotely. In order to address such limitations, this study proposes a framework for estimating flood water depths and enhancing flood delineations, based on readily available topographical data. The methodology is specifically designed to accommodate, as additional inputs, masks delineating water bodies and/or no-data areas. In particular, the method relies on simple morphological arguments to expand flooded areas into no-data regions, and to estimate water depths based on the terrain elevation of the boundaries between flooded and non-flooded areas. The underlying algorithm – named FLEXTH – is provided as Python code and is designed to run in an unsupervised mode in a reasonable time over areas of several hundred thousand square kilometers. This new tool aims to quantify and ultimately to reduce the impacts of floods, especially when used in synergy with the recently released Global Flood Monitoring product of the Copernicus Emergency Management Service.

## 1 Introduction

Floods are among the most devastating of natural disasters, causing widespread destruction and loss of life across the globe (EMDAT, 2022; Douris and Kim, 2021). Accurate and timely flood mapping is essential for effective disaster management,

facilitating early warnings, evacuation planning, proactive response and subsequent recovery (Voigt et al., 2016). In the past,
flood mapping relied heavily on ground-based observations, which often proved insufficient for real-time monitoring due to
limitations in spatial and temporal coverage. The advent of satellite remote sensing technology and advancements in data
processing techniques have revolutionized flood mapping, offering substantial benefits in terms of accuracy, coverage, and
timeliness of information delivery (Schumann and Moller, 2015; Salamon et al., 2021). With satellites, floods can be monitored
remotely and continuously over vast and inaccessible areas. These aspects are especially relevant for vulnerable regions where
risk mitigation strategies are lacking and the response to disasters is often inadequate. The data obtained from satellites provides
valuable insights into flood dynamics, such as water extent and the progression of inundations over time (Spasova and Nedkov,
2019). This information is crucial for emergency response planning and post event recovery, ultimately contributing to saving
human lives and mitigating social and economic impacts.

Among the satellite-borne sensors, optical sensors (i.e. those operating in the optical range of the electromagnetic spectrum)
have long been used to map floods (e.g. Grimaldi et al. (2016)). However, despite the extensive coverage of optical satellite
imagery, it is still challenging to map floods from space under certain circumstances, either in a supervised or an unsuper-
vised mode. For example, optical sensors can not "see" at nighttime or through clouds. The latter limitations are especially
problematic, since adverse meteorological conditions may be expected during extreme hydrological events.

More recently, Synthetic Aperture Radar sensors (SAR) have offered an effective alternative for flood mapping, thanks to
their high capacity to discriminate surface waters (Clement et al., 2018; Jo et al., 2018; Pulvirenti et al., 2011). Operating in
C-band microwave frequencies (i.e. between 4 and 8 Ghz), SAR sensors such as the one on board the Copernicus Sentinel-
1 satellites, allow for an efficient and comprehensive assessment of floods worldwide, day and night and regardless of the
weather conditions. Despite these advantages, the efficacy of SAR sensors can be limited in certain situations, including: i)
low sensitivity, where dry/wet areas may be misclassified due to the presence of dense vegetation or urban areas; ii) water
look-alike conditions, where the ground surface interacts with the incoming radar signal as if it were water (e.g. smooth or very
dry surfaces, or wet snow).

Masking areas where flood mapping is not feasible, is not only a matter of scientific realism, but ultimately increases the trust
of users in the flood delineation products. In this context, one of the "output layers" provided by the Copernicus Emergency
Management Service's recently released Global Flood Monitoring (GFM), is an "Exclusion Mask" which excludes areas that
challenge SAR-based flood mapping (https://emergency.copernicus.eu/). In practice, the mask is equivalent to no-data in areas
where the system is unable to confidently discriminate floodwaters.

GFM is an online system that provides, as its main output, worldwide flood delineations by automatically ingesting and
processing in near-real-time all incoming Sentinel-1 SAR acquisitions. As part of the GFM methodology, as soon as a new
Sentinel-1 image is available, the raw SAR backscatter data is promptly processed by three separate state-of-the-art flood
classification algorithms, in an unsupervised manner. The final flood map is generated via an ensemble approach that increases
the robustness and reliability of the final products (Salamon et al., 2021; Krullikowski et al., 2023), together with a series of
additional output layers (including the Exclusion Mask).

Water depth is considered to be the most informative proxy variable for quantifying flood impacts. In fact, empirical depth-damage curves have traditionally been designed to quantify economical losses as a function of water depth for different exposed assets in different regions (Jongman et al., 2012; Huizinga et al., 2017). Despite the best practices and most recent advances, water depth cannot be estimated by satellite-based flood mapping. Water depth estimation and the accurate identification of flood extent are critical tasks, not only for disaster risk management, but also for other scientific disciplines, including geomorphology, hydrology and climate change analysis (e.g. Feyen et al. (2020); Rossi et al. (2023)). Traditional approaches for estimating water depth and flooded area rely on ground-based observations and manual measurements, which suffer from limitations such as time-consuming data collection and limited spatial coverage. Remote sensing technologies and digital terrain models (DTMs) have opened promising avenues for addressing these challenges (Fuentes et al., 2019; Khattab et al., 2017). For example, Cohen et al. (2018) developed an effective and widely used framework to compute water depth from the intersection of flood delineation polygons and a DTM (Teng et al., 2022; Penton et al., 2023). A second release of the methodology improved some of the limitations of the first release, such as the computational inefficiency and the impossibility to properly consider the boundaries between floods and permanent water bodies (Cohen et al., 2019). However, the framework still suffers from substantial processing times over very large areas (or with high resolution rasters) and can provide unrealistic water levels and depths in certain conditions (Cohen et al., 2019). Other approaches for estimating flood depths at a local scale have proved effective (Bryant et al., 2022; Cian et al., 2018). Nevertheless, their applicability for large-scale assessments has not yet been demonstrated. Furthermore, these approaches are rather elaborate, and are unlikely to be suitable for unsupervised use over large areas. Finally, none of the current methodologies takes advantage of topographical information to enhance flood delineations in a non-trivial manner (i.e. other than with so-called "bathtub model" approaches), nor have they been tested in complex riverine environments with irregular topographies, or they employ closed-source/commercial softwares (Rodriguez Enriquez et al., 2023).

To address the aforementioned limitations, the present study introduces a novel algorithm called FLEXTH. The special feature of FLEXTH is its utilization of topographic information to effectively handle flood maps with gaps arising from areas with no-data or seasonal/permanent water bodies. FLEXTH aims to expand flood delineations and provide estimates of water level and water depth seamlessly across the entire study area. The combined use of satellite-derived inundation maps and DTMs not only improves the overall accuracy and reliability of the flood assessment, but also enhances the ability to model and predict flood dynamics in areas prone to inundation. Furthermore, the framework is suitable for any flooding mechanism, namely riverine, coastal and pluvial. Other key features of the methods described in this paper are: i) a limited requirement for supervision since FLEXTH is designed to operate automatically over large areas; ii) computational efficiency, which is achieved using computer vision algorithms that entail reasonable processing times, for areas in the order of up to $10^9$ pixels.

The workflow that is presented in this study provides a comprehensive and efficient approach for enhancing (satellite-based) flood maps, and for complementing flood extent information with estimates of water level and water depth, with potential improvements to flood assessment, impact calculation and disaster response strategies.

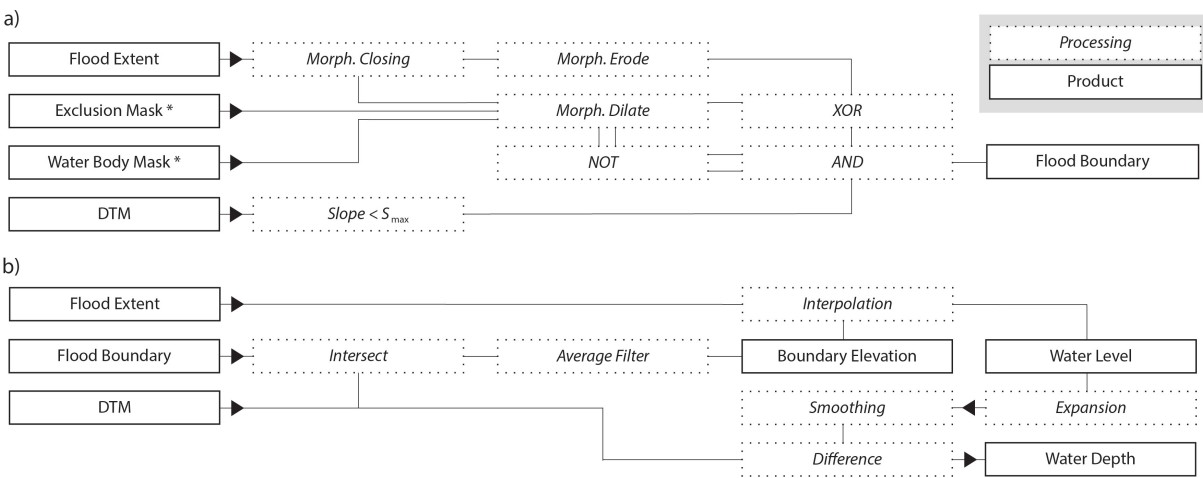

**Figure 1.** Workflow conceptualizing the main processing steps and the final/intermediate products: a) identification of valid wet-dry boundaries of flooded areas; b) computation of the water level and water depth in the initial and expanded flooded areas. Asterisks denote optional inputs, see Section 2.1.

## 2 Methods

The required and optional input data for the workflow illustrated in Figure 1 are introduced in Section 2.1. Section 2.2 describes the procedure that estimates water level and water depth in areas initially delineated as flooded. Section 2.3 describes the routine that propagates the flood water inside the no-data areas by taking advantage of a DTM. Note that this final aspect is only relevant if a no-data mask is provided as input.

The underlying algorithm –named FLEXTH– is available as an open access Python script: see the code availability section. An additional script is provided for easily preprocessing the input data (i.e. cropping, resampling and reprojecting the DTM to the same grid as that of the flood delineation raster).

### 2.1 Input and Output products

The processing chain has, as a minimum requirement, two inputs: i) a binary raster map delineating the flooded areas; ii) a DTM (Figure 1). Additionally, users can take advantage of two optional input data layers: iii) a binary no-data mask delineating all areas excluded from flood mapping); iv) a binary map identifying permanent and seasonal water bodies. The no-data mask is particularly relevant when a satellite-borne sensor is unable to discriminate reliably flooded areas due to, for example, low sensitivity or water-look-alike conditions. A no-data mask may correspond to clouds in the case of optical image data (e.g. from Sentinel-2), or to densely vegetated/urbanized areas in the case of SAR sensors (e.g. Sentinel-1).

All processing steps are computed on georeferenced arrays (geotiff) with matching pixels: i.e. the inputs share the same spatial extent, pixel size and projected reference system. As outputs, FLEXTH delivers water level and water depth maps, possibly expanding throughout no-data areas, if provided.

## 2.2 Water level and water depth estimation

Water level (WL) refers to the elevation of the water surface above an arbitrary vertical datum, while water depth (WD) is the difference between the elevation of the water surface and the underlying terrain. The idea behind the water level estimation method presented here, is that within each contiguous (or connected) flooded area, the water level can be inferred based on the ground elevation along the corresponding wet-dry boundary. In particular, along the borders of flooded areas, the water level must fall in the interval between the elevation of neighboring flooded/non-flooded pixels, see Figure 2a. The elevation of the

terrain around flooded areas, including neighboring flooded (i.e. wet) and non-flooded (i.e. dry) pixels, is therefore critical. The following steps describe how to link the wet-dry boundaries (obtained from flood inundation maps) with topographical information (from DTMs) in order to estimate water levels inside contiguous flooded areas.

The workflow, as illustrated in detail in Figure 2, consists of three main phases. The first phase, shown in Figure 2c, focuses on identifying the reference outline of each contiguous flooded area. The second phase, uses the reference outline in combina-

120 tion with a DTM to compute water level and water depth inside the flooded areas, Figure 2b. The third phase employs the water level estimates and topographical information to propagate flood water across hydraulically-connected no-data regions, Figure 2e. On a computational level, the procedure takes advantage of the Open Source Computer Vision Library (https://opencv.org/) which was developed to facilitate efficient image processing operations over large raster datasets (Bradski, 2000).

As a pre-processing step (Figure 1a), two rounds of "morphological" closing using a 3x3 cross kernel are performed on the

125 initial flood delineation (Gonzalez and Woods, 2018; Haralick and Shapiro, 1992). This procedure can be relevant especially for automatically derived flood maps, where flooded areas may feature small inaccuracies and noisy borders. In fact, the procedure regularizes the wet-dry contour and fills small gaps that could otherwise negatively affect the following steps.

At this point, the borders of flooded areas, including neighboring wet and dry pixels, are identified by subtracting (logical XOR) a morphological erosion of the flood map from its morphological dilation (Gonzalez and Woods, 2018). Both operations

are performed using a 3x3 box kernel (see Figure 2b, steps i to iv).

However, flooded areas may not simply be enclosed by "valid" wet-dry boundaries, but they can also share borders with no-data regions or with permanent water bodies. In the first case, the true location of the wet-dry divide is unknown, as it may be hidden inside the "blind" no-data areas. In the second case, when a flood merges with a water body, the shared wet-wet boundary doesn't provide any topographic information useful for determining the water level. For the identification of reliable

water levels, it is therefore crucial to consider just the informative borders, and to exclude the spurious ones. For this purpose, a dilation of both no-data areas and water bodies (with a 3x3 kernel) is used to mask the conterminous wet-dry contours identified previously (Figure 2c, steps iv to vi). This operation correspond to combining with a logical AND the initial outline of flooded areas with the complement (i.e. logical NOT) of the dilated exclusion and water bodies mask, as per Figure 1a.

Additionally, border pixels corresponding to topographic gradients larger than the user-defined threshold $S_{max}$ are excluded

from the following computations (Figure 1a). In fact, large topographic gradients along the wet-dry boundaries, especially in combination with coarse resolutions of the input flood map and/or DTM, can lead to erroneous estimates of representative water levels (Cohen et al., 2022).

Because water level ideally lies between the elevations of neighboring wet-dry pixels (Figure 2a), the DTM values corresponding to the remaining valid contour pixels are processed with a moving average filter (3x3 box kernel) in order to compute the representative water level along the outlines of the flooded areas (red dashed boxes in panel vii-viii of Figure 2d, see also Figure 2b).

The representative water level corresponding to the flooded/non-flooded boundaries is now used to extrapolate the water level inside each flooded area (step ix in Figure 2d). Contiguous flooded areas are identified by means of a connected component analysis and processed independently (Stockman and Shapiro, 2001).

The water level of each flooded pixel can then be estimated based on two alternative methodologies. Method A estimates water level at a target pixel as the distance-weighted arithmetic mean of the $N_{max}$ closest pixels (in the Euclidean sense) belonging to the border of the corresponding flooded area. The weight $w = 1/d^\alpha$ is controlled by the parameter $\alpha$ which modulates the range of influence of each pixel of the border having a distance $d$ from the target location. Alternatively, Method B requires the selection of a percentile $P$ of the distance-weighted distribution of the elevations along the border pixels as a reference for assigning water levels. Inverse-distance weighted percentiles are computed by scaling the frequency of the elevation of each pixel along the border by a factor $1/d^\alpha$. In general, Method A tends to provide smoother estimates of water levels. However, Method B can be more robust and versatile when the altimetry of the cells along the border is biased or is poorly representative of the actual water level. This may be due, for example, to inaccuracies in the delineation of flooded areas and/or errors in the DTM, or in case of coarse raster resolutions (Cohen et al., 2022).

If the number of pixels along the wet-dry boundary is too small to provide robust estimates of the water level (say they are less than $N_{min}$), an alternative approach is adopted. Specifically, in these cases, the water level is set as the $P_{in}$ percentile of the distribution of DTM values inside that flooded region. Such an alternative approach is more likely to be required in the presence of small flooded areas located in steep topographies surrounded by extensive water bodies and/or no-data areas.

Finally, where the estimated water level is lower than the ground elevation, a fictive water depth $WD^*$ is assigned. For consistency, the same $WD^*$ is added to the remaining water depth estimates. Water level estimates lower than ground elevation can occasionally occur in practice as a result of, for example: i) imprecise flood delineation; ii) inaccuracies in the DTM; iii) vertical curvatures of the water surface due to hydrodynamic effects.

## 2.3 Flood expansion

Once water levels are computed in the delineated flooded areas, water is recursively spread from flooded locations into neighboring masked (i.e. no-data) areas, provided that the ground elevation of the no-data pixel is lower than the water level in the nearby source pixel (see Figure 2e, step x). To prevent unlimited and unrealistic spreading of flood water, a routine regulates water levels and allows water to spread up to a maximum distance (computed from the point along the initial flood boundary where flood propagation began). In fact, without a constraint on flood spreading – such as in the case of a horizontal water level propagation as in a bathtub filling approach (e.g. Nobre et al. (2016)) – flood water could propagate indefinitely. Such a risk may especially arise when extensive excluded areas are combined with regional topographical gradients or in large-scale assessments.

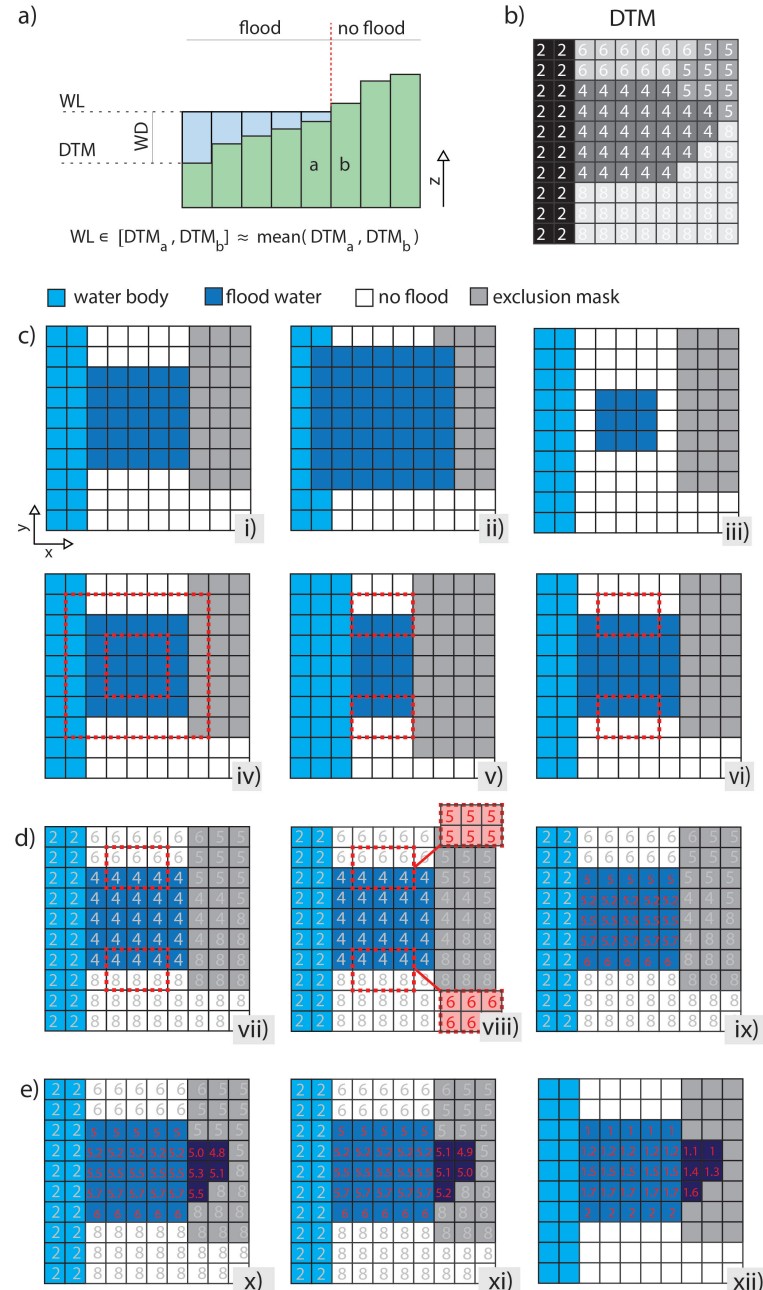

**Figure 2.** Schematic representation of the main steps of the procedure to estimate water depths and to expand floods inside excluded areas: a) principle behind the water depth estimation combining a flood map and a DTM; b) sample DTM with elevation values in each cell; c) identification of valid wet-dry boundaries of a flooded area; d) water level and water depth estimation inside the initially flooded areas; e) flood propagation in the no-data mask. See Section 2.2 and 2.3 for details.

A maximum propagation distance $d_{max}$ is assigned to each contiguous flooded area as a function of its extension $A$ (km$^2$), allowing large flooded areas potentially to spread further. Specifically, flood expansion is controlled via the exponential relationship:

$$d_{max}(A) = D_{max}\left(1 - 2^{-\frac{A}{A_{1/2}}}\right) \tag{1}$$

where $D_{max}$ (km) corresponds to the maximum allowable propagation distance for an arbitrarily large flooded area, and $A_{1/2}$ (km$^2$) establishes the flood extent for which $d_{max} = D_{max}/2$. This simple parameterization is heuristically rooted in the mass conservation principle: large flooded areas contain more water that can potentially propagate further. However, a sill is enforced as – in reality – flood propagation cannot increase indefinitely.

To improve computational speed, flood propagation follows a forward explicit scheme which ensures a progressive decrease of water depth as the flood spreads. The numerical method begins by propagating water levels to neighboring excluded pixels starting from the contours of each flooded area, and it continues until a set of conditions is met. In particular, starting from a generic pixel at position $i, j$, the algorithm estimates water levels in neighboring excluded pixels at position $i \pm 1, j \pm 1$ ($WL_{i \pm 1, j \pm 1}$) based on the ground elevation $DTM_{i \pm 1, j \pm 1}$, the initial water level $WL_0$ at the origin of the propagation, and the distance $d_{i \pm 1, j \pm 1}$ computed along the propagation route until position $i \pm 1, j \pm 1$. Valid pixels where flood propagation is performed are those fulfilling all the following conditions: i) belong to the no-data mask; ii) $DTM_{i \pm 1, j \pm 1} < WL_{i,j}$; iii) $WL_{i \pm 1, j \pm 1} < WL_{i,j}$; iv) no flood propagation has yet been performed on the target location. All the eight pixels surrounding each starting location are considered as neighbors (8-connectivity). Specifically, the routine follows the expression:

$$WL_{i \pm 1, j \pm 1} = WL_0 - (WL_0 - DTM_{i,j})\frac{d_{i \pm 1, j \pm 1}}{d_{max}} \tag{2}$$

Figure 3 illustrates the behavior of the flood propagation routine in a few one-dimensional sample cases. Specifically, the Figure shows how an initial water depth $WD_0$ propagates on different land surface topographies characterized by a starting value $DTM_0$ and a combination of a linear, a sinusoidal and a random trend parametrized via $s_i$, $s_s$ and $s_r$ as:

$$DTM(x) = DTM_0 + s_i x + s_s \sin(2\pi x) + s_r z_{rand} \tag{3}$$

where $x$ is a longitudinal dimension (in kilometers) and $z_{rand}$ is a uniformly distributed random number in the interval $[-0.5, 0.5]$ (see caption of Figure 3 for details).

As a final post-processing step (Figure 2e, panel xi), water level estimates in the expanded regions are smoothed in order to: i) reduce potential discontinuities between the elevation of the water surface and the ground elevation in contiguous non-flooded pixels; ii) homogenize water levels where floodwaters spreading from different sources merge together. For this purpose, a 5x5 circular convolutional smoothing filter is slid twenty times over $WL'$, i.e. the union between the estimated water level map and the DTM in non-flooded areas. Specifically, $WL' = WL(flood = 1) \cup DTM(flood = 0)$, where $WL$ can in turn be seen as

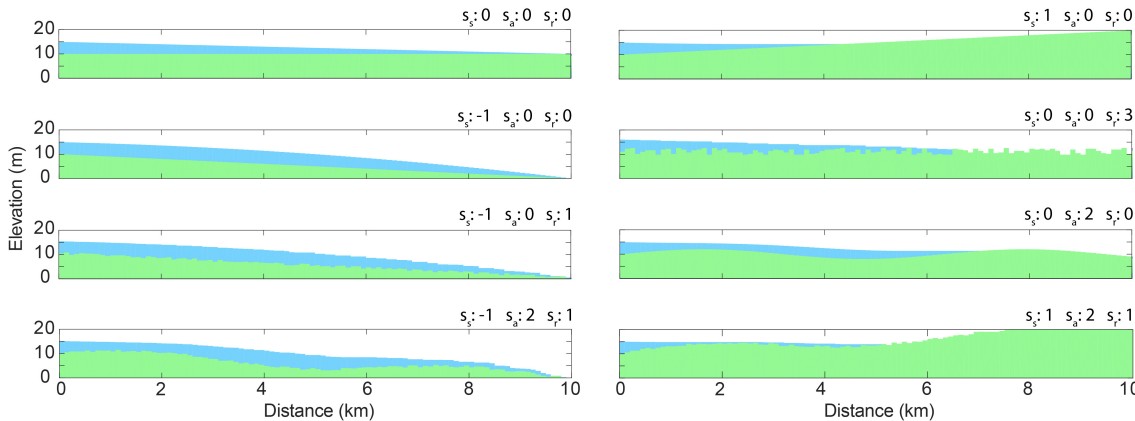

**Figure 3.** One-dimensional examples of the flood expansion routine over synthetic land surfaces featuring different topographies. Topographies are parameterized combining a linear trend $s_i$ (‰), a sinusoidal component with amplitude $s_s$ (m) and period $2\pi$, and a random component ($z_{rand} \sim s_r\mathcal{U}(-0.5, 0.5)$ with $s_r$ (m) as scale). Flood water propagates from left to right, starting from $WD_0 = 5\,m$ and $DTM_0 = 10\,m$ until the maximum assigned propagation distance $d_{max} = 10\,km$. Note that $d_{max}$ can be reached just under some topographical conditions.

the union between water levels estimated inside the initially flooded areas ($WL_0$) and the water levels in the expanded flooded areas ($WL_e$). At each recursive application of the filter, the initial values of $DTM(flood = 0)$ and $WL_0$ are reestablished, ensuring this post-processing step to be limited to $WL_e$. The procedure terminates with the step xii of Figure 2e, where water depth is estimated as the difference between water level and the DTM.

## 3 Results: the Pakistan 2022 case study

The framework described in Section 2 is tested for the use case of the devastating flood that hit the Indus valley in Pakistan between July and September 2022. The flood caused severe destruction, affected about 33 million peoples, claiming over 1700 victims and causing about 20 billion euros of direct damages (Nanditha et al., 2022). Given the severity and the wide extent of the event – which was by far the world's largest flood during the period of the current study – it is well suited as a proof of concept for the methodology introduced in Section 2.

Specifically, the procedure described in the previous section is coupled with the maximum flood extent derived by aggregating over the months of July, August and September all flood delineations provided by GFM over an area of $2.43 \cdot 10^6$ km$^2$, resulting in about $10^8$ flooded pixels at 20 m resolution. The exclusion mask and water bodies are also based on GFM, whereas topographic information are provided by FABDEM, a recently released DTM whose goal is to remove vegetation and buildings from the Copernicus DEM by using AI and ancillary satellite data (ESA, 2022; Hawker et al., 2022).

Figure 4 displays the flooded area and the corresponding water depth computed as described in Section 2 over the most severely affected region. The result – featuring the morphology-based flood expansion – covers a total flooded area of 61331 km$^2$ (including neighboring flooded areas not shown in the figure), which is about 50 % larger than the initial GFM-based flood

**Table 1.** Parametrization of the workflow used in the case study. Water depths and water levels are estimated using Method A (see Section 2).

| Parameter | Description | Value | Units |
|-----------|-------------|-------|-------|
| $S_{max}$ | Maximum admissible slope for the wet-dry reference border pixels | 0.1 | - |
| $N_{max}$ | Maximum number of border pixels used to estimate WL | 100 | - |
| $N_{min}$ | Minimum number of valid border pixels required to estimate WL based on the default procedure | 10 | - |
| $P_{in}$ | Percentile of the distribution of ground elevation underneath a flooded area used as a reference to estimate WL (alternative procedure) | 0.98 | - |
| $\alpha$ | Inverse distance weighting rate for WL interpolation within flooded areas | 2 | - |
| $WD^*$ | Additional water depth assigned to account for WL estimates lower than ground elevation | 0.1 | m |
| $D_{max}$ | Maximum propagation distance for an arbitrarily large flooded area | 10 | km |
| $A_{1/2}$ | Initial size of a flooded area for which $d_{max} = D_{max}/2$ | 100 | km$^2$ |

delineation (39333 km$^2$). The results correspond to the parametrization summarized in Table 1. The sensitivity of the proposed methodology on model parameters is discussed in Section 6.

For the use case, computations are run in parallel on batches of ten of the native 300x300 km tiles constituting the tiling system of GFM (see: Bauer-Marschallinger et al. (2014) and https://extwiki.eodc.eu/GFM/PDD). The full running time featuring flood expansion and water depth estimation required to generate the result displayed in Figure 4 is about 1.5 hours on a dual 6 Core CPU workstation (Intel Xeon C5-2620 v2) with 44 Gb of RAM.

### 3.1 Evaluation of the flood extent and water depth estimates

In this section, the performance of the framework presented in Section 2 is evaluated quantitatively. In particular, subsection 3.1.1 and 3.1.2 focus on evaluating the flood extent and the flood depth respectively. It is worth noting that ground truth derived via observations acquired during field surveys are rarely available for flood extents, and are even scarcer for water level and depth– especially at the scales considered in this study. Furthermore, despite the long timescale of the event, reference data might display flood conditions at different stages.

#### 3.1.1 Evaluation of the flood extent

The accuracy of the flood extent is evaluated using, as a reference, the flood maps produced by the Copernicus Emergency Management Service (CEMS) corresponding to the three AOIs delineated in Figure 4 (EMSR629 activation, https://emergency. copernicus.eu/mapping/list-of-components/EMSR629). The maps cover 3000 km$^2$ in total, and aim to capture the maximum flood extent around the cities of Larkana (AOI 1), Shikarpur (AOI 2) and Jacobabad (AOI 3). The CEMS maps in vector format were produced in a semi-automatic/semi-supervised way with expert knowledge refinement starting from the imagery acquired by the SPOT6/7 sensor on the 30[th] of August 2022 (Roth et al., 2022).

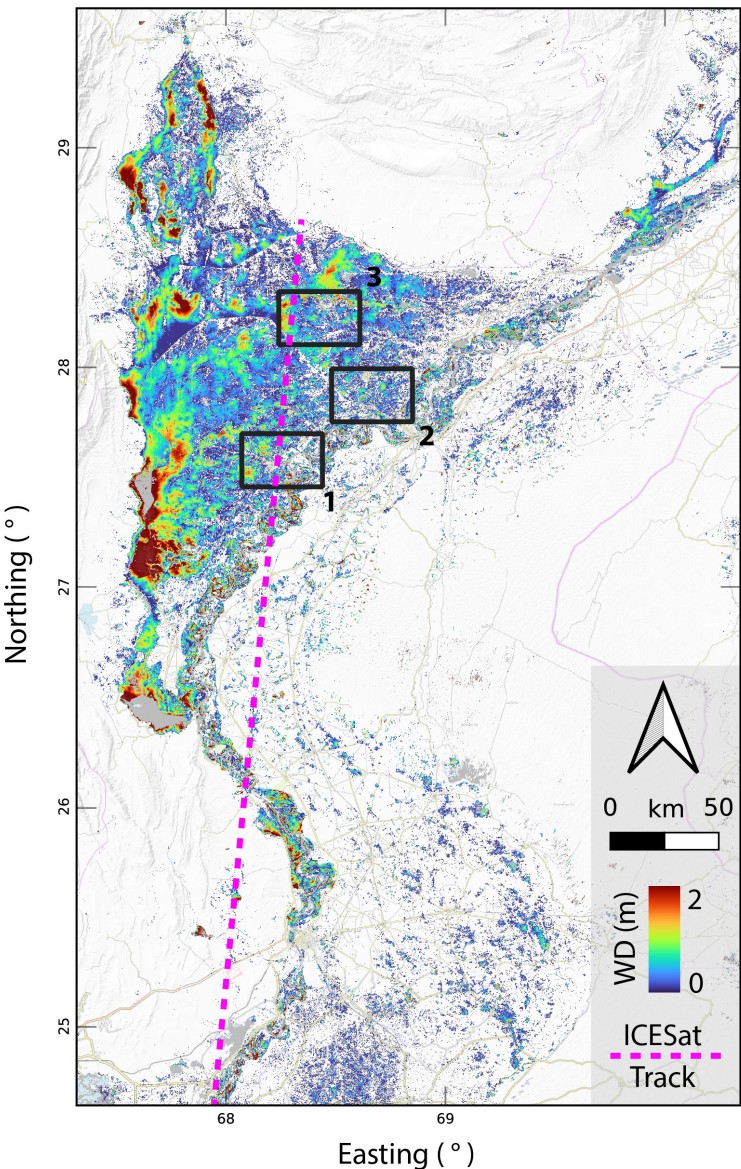

**Figure 4.** Flood delineation featuring flood expansion and water depth calculation as described in Section 2 over the most affected area along the Indus valley (Pakistan) during the flood of July-September 2022. Permanent and seasonal water bodies are depicted in gray. The three areas of interest (AOIs) are also highlighted.

Figure 5 compares the CEMS flood delineation against the flood classification obtained by the temporal merging of the GFM flood extent layers (first row), as well as the GFM-based flood expansion procedure performed as described in Section 2 (second row). For completeness, the third and the fourth rows of Figure 5 displays the water level and the water depth estimated

**Table 2.** Classification accuracy metrics of the extended flood maps displayed in Figure 5: Critical Success Index (CSI); $F_1$ score, User Accuracy (UA) and Producer Accuracy (PA) with respect to the "flood" class; relative frequency of True Positive (TP), True Negatives (TN), False Positives (FP) and False Negatives (FN). The subscripts are the metrics for the GFM flood delineation without flood expansion (i.e. first column in Figure 5). In parentheses is the percent change of the extended flood maps relatively to the case without flood expansion.

| | CSI (%) | $F_1$ (%) | UA (%) | PA (%) | TP (%) | TN (%) | FP (%) | FN (%) |
|---|---|---|---|---|---|---|---|---|
| AOI 1 | $69_{65}$ (6) | $82_{79}$ (4) | $77_{81}$ (-5) | $88_{77}$ (14) | $51_{44}$ (16) | $26_{31}$ (-16) | $16_{11}$ (45) | $7_{13}$ (-46) |
| AOI 2 | $68_{58}$ (17) | $82_{73}$ (12) | $85_{88}$ (-3) | $76_{63}$ (21) | $54_{43}$ (26) | $21_{24}$ (-13) | $9_6$ (50) | $15_{26}$ (-42) |
| AOI 3 | $63_{63}$ (0) | $77_{77}$ (0) | $65_{68}$ (-4) | $94_{89}$ (6) | $52_{48}$ (8) | $18_{23}$ (-22) | $28_{22}$ (27) | $3_6$ (-50) |

as described in Section 2. Table 2 summarizes the performances of the classification for the two different scenarios (i.e. with and without flood expansion).

### 3.1.2 Evaluation of water depth estimates with ICESat-2

This section describes the approach developed to assess the water depth estimates computed in Section 2, using as a benchmark the altimetric data acquired by the ICESat-2 satellite mission of NASA (https://icesat-2.gsfc.nasa.gov). ICESat-2 features ATLAS, an accurate laser altimeter designed for worldwide recording along six parallel tracks of ground points at a sampling frequency of up to about 1 point/m and a revisit time of 91 days (Neuenschwander and Pitts, 2019; Dandabathula et al., 2023; Wang et al., 2019; Li et al., 2021).

ICESat-2/ATLAS delivers twenty-two standard products (i.e. ATL00 - ATL21) divided into four levels (i.e. levels 0 - 3) characterized by increasing levels of processing. The current analyses employ ATL03, a Level 2 product that maintains the full altimetric information after basic post-processing steps on the initial raw telemetry data (Zhu et al., 2022). ICESat-2 acquires altimetric data along three pairs of tracks, each pair being characterized by a weak and strong beam of photons (4:1 energy ratio). Pairs of tracks are 3.3 km apart in the across-track direction, while strong and weak beams have a transversal offset of 90 m. For the analysis, only medium and high confidence photons from the strong beams are used.

On the 3 September 2022, ICESat-2 acquired data along a 500 km track through the study area (see Figure 4). The timing of the acquisition is compatible with the maximum flood extent in the area, estimated to be around the end of August (Nanditha et al., 2022; Roth et al., 2022). Such data are used as a reference altimetry during the flood. Furthermore, two of the three strong tracks acquired by ICESat-2 on 3 September have a good spatial match (i.e. they are about 50 m apart in the across-track direction) with two of the strong tracks acquired by ICESat-2 on the 6 June 2021. Each pair of spatially matching ICESat-2 tracks during the flood period (September) and during the dry period (June) are denoted as $I_{wet}^i$ and $I_{dry}^i$ respectively, with $i = 1, 2$ denoting the two spatially matching tracks and $I_* = I_*^1 \cup I_*^2$. The wet (dry) tracks feature a total of about 1 (0.8) million points, with an average spacing of 0.31 (0.62) m and similar inter-quartile ranges of about 0.71 m for both tracks.

The altimetric data acquired by ICESat-2 during the dry and wet conditions are used to assess water depth estimates computed as described in Section 2. In particular, $I_{dry}$ serves as a reference ground topography, whereas $I_{wet}$ is expected to be positively biased by flood water.

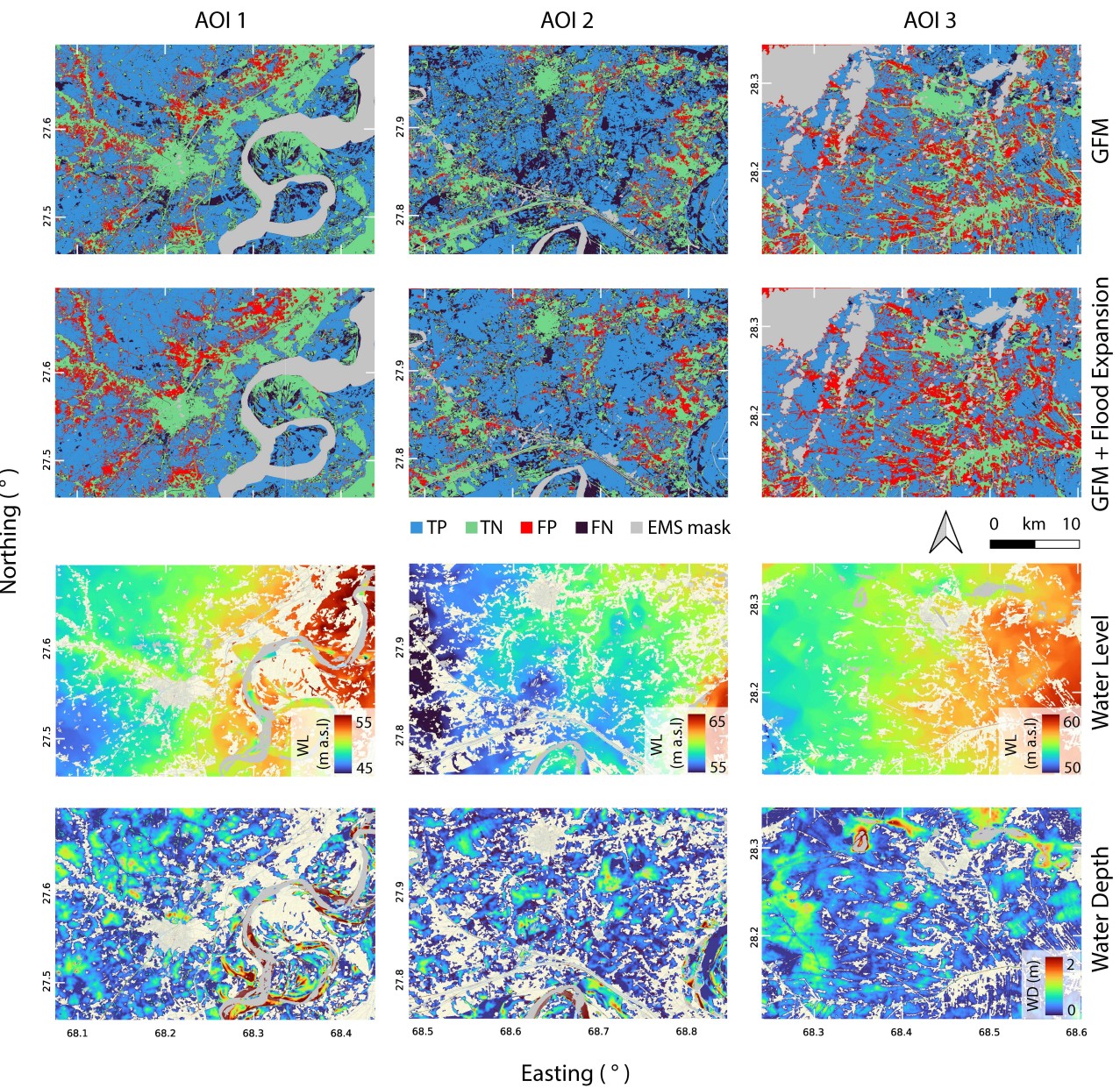

**Figure 5.** Classification results of: i) aggregated flood delineations obtained by merging GFM products over the months of July, August and September 2022 (first row), and ii) the flood expansion procedure described in Section 2, (second row) for the 3 AOIs considered. Classification performances are displayed in terms of true positives (TP), true negatives (TN), false positives (FP) and false negatives (FN) with respect to the reference CEMS rapid mapping products. Panels in the third and fourth rows: water level and water depth computed following the procedure outlined in Section 2.

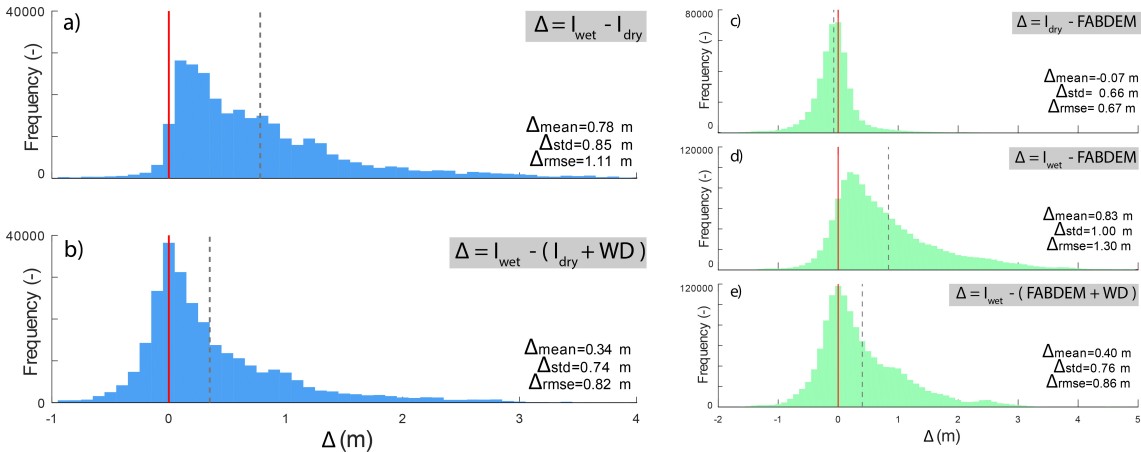

**Figure 6.** Distribution of the difference between altimetric data acquired by ICESat-2 during wet and dry conditions along a 500 km track across the study area with (or without) considering water depth estimates (panels a and b). Comparison of the ICESat-2 altimetric data with FABDEM in wet and dry conditions including (or not) water depth estimates (panels c d and f). The red vertical line marks the ideal condition, where the water depth reconstructed by FLEXTH perfectly matches the one computed with ICESat-2. The dashed line corresponds to the mean of the distribution.

As a preparatory step for the analyses, the $I_{dry}$ and $I_{wet}$ tracks are processed with a moving median filter (with a window over 100 photons) in order to minimize the noise which characterizes the data and to reduce potential biases caused by vegetation and/or buildings. The filtered photons – which are irregularly spaced along the tracks – are then resampled on the same $5 \cdot 10^{-5}$ degrees latitude intervals (about 5 m). As ICESat-2 follows a north-south orbit, altimetric observation at resampled locations correspond to the same latitudes, while longitudinal differences are due to the ~50m offset between $I_{dry}$ and $I_{wet}$. As a result, resampled data from the dry and wet tracks represent surface elevation at approximately the same locations.

Considering the water depths computed as described in Section 2 and sampled in correspondence of the $I_{wet}$ photons, Figure 6a and b display the distribution of the difference $\Delta = I_{wet} - I_{dry}$ and $\Delta = I_{wet} - (I_{dry} + WD)$ for all flooded pixels delineated in Figure 4. The distribution of the elevation differences highlights that including water depth estimates based on the proposed procedure substantially reduces both the bias and the variability in the distribution of $\Delta$. Furthermore, the mode of the distribution correctly matches 0 when water depth estimates are added to the ground elevation measured by ICESat-2 during dry conditions.

As a complement, the distributions in panels c d and e of Figure 6 compare ICESat-2 altimetry both in absence of flooding ($I_{dry}$) and in flood condition ($I_{wet}$) with FABDEM. Panel c shows that along the sampled points FABDEM has a small positive bias compared to ICESat-2 and a fairly symmetrical distribution during dry conditions. On the other hand, as expected, during the flood the distribution shifts towards larger values of $I_{wet}$ with a substantial skewness in the distribution. Adding the water depth estimates effectively reduces the bias (panel e), but underestimated flood depths remain.

## 4 Evaluation of FLEXTH-derived water depth and water level estimates against hydrodynamic simulations

This section compares the water level and water depth computed by FLEXTH with the results of hydrodynamic simulations in two use cases: i) along a 40 km stretch of the Brazos river near Huston, Texas (30.18°N; 96.18°W); ii) along a 150 km river network corresponding to the confluence of the rivers Tera, Orbigo and Esla in north-western Spain (41.95°N; 5.67°W). Dedicated simulations are performed for the Brazos river case, whereas the national flood hazard maps are used for Spain. Section 4.1 describes the benchmark data.

The hydrodynamic simulations do not aim to reproduce specific observed events. Yet, they do offer physically-based and realistic scenarios that can effectively serve as benchmarks for the methods presented in this study. In fact, simulations readily provide water level and water depth estimates, as well as a clear delineation of the flood extent, bypassing some of the limitations of remote-based flood mapping (see Section 1).

FLEXTH is also compared against the latest version of FwDET-GEE-v2, the Google Earth Engine implementation of FwDET (Cohen et al., 2022, 2019). FwDET-GEE-v2 is an updated and improved version of FwDET-GEE (Peter et al., 2020) and is a widely used tool to rapidly estimate flood water depth based on flood delineations and terrain topography (e.g. Penton et al. (2023), see also Section 1).

The water depth and water level estimated by FLEXTH and FwDET are obtained providing the two algorithms with the same inputs, namely: the topography of the area, and a binary map covering all areas denoted as flooded as per the reference hydrodynamic model. As no-data masks are unavailable for the two sites, the flood propagation routine of FLEXTH is not performed (see Section 2.3).

### 4.1 Hydrodynamic benchmark data

### 4.1.1 Brazos river – USA

Water level and water depth are simulated in steady-state conditions across an unstructured triangular mesh with $4.5 \cdot 10^6$ elements (average element size of 50 m$^2$) with the freeware software Basement 4.0.2 HPC$^©$ (https://basement.ethz.ch). Basement solves the 2-D shallow water equations across the flow domain using a finite volume approach under a set of boundary conditions (Vanzo et al., 2021). In the simulation, the model is forced with the 4500 m$^3$/s peak discharge recorded at the USGS gage near Hempstead (ID: 08111500) during the May 2016 flood event (Zhang et al., 2018). The input discharge is homogeneously distributed along the upstream section, imposing uniform flow with a 0.5 ‰ hydraulic gradient. The same 0.5 ‰ hydraulic gradient is assigned at the outlet section. Input and output sections span across the entire floodplain, which is visible from the topography of the area (see Figure 7c). Two Strickler roughness coefficients are assigned to the flow domain: 35 m$^{1/3}$s$^{-1}$ in the main channel and 20 m$^{1/3}$s$^{-1}$ in the floodplain (Chow, 2009). The morphological characterization of the area is acquired from the latest release of USGS 3D Elevation Program (3DEP) Datasets at 1/3 arc-seconds (approximately 10 m) available at www.sciencebase.gov/catalog. Note that, although a recorded discharge is used as an input for the simulation, roughness coefficients are not calibrated. In fact, the goal of the simulation is not to accurately reproduce an observed event, but is to provide a virtual – yet realistic – scenario to be used for benchmarking purposes. The simulation resulted in a flooded area of 103 km$^2$

with an average water depth of 1.34 (standard deviation: 2.41 m). Ultimately, water level and water depth are resampled from the native Basement's triangular mesh to a regular 10 m grid, resulting in $1.04 \cdot 10^6$ flooded pixels.

### 4.1.2 Tera, Orbigo and Esla confluence – Spain

In this case, the water depth is provided by the official Spanish flood hazard maps at 100 years return period, openly available at https://centrodedescargas.cnig.es/CentroDescargas. The maps are steady-state numerical solution of the 2D shallow water equation obtained with the software InfoWorks RS$^{©}$. The simulations are performed based on a 1 m hydraulically-conditioned DTM with all major artificial structures removed from the floodplain, while the final results are provided at 2 m resolution. The extensive technical documentation underlying the simulations is accessible in Sánchez and Lastra (2011). As water level is not provided, it is derived by adding the topography to the water depth estimates. For this purpose, a 2 m lidar-derived DTM of the area was used (https://centrodedescargas.cnig.es/CentroDescargas).

The flood covers 256 km$^2$ ($63.97 \cdot 10^6$ pixels) with an average water depth of 1.29 m (standard deviation: 1.46 m).

### 4.2 Results: FLEXTH Vs Hydrodynamic simulations

The results delivered by FLEXTH in this section are obtained using the same parametrization as in Pakistan (see Section 3 and Table 1). FwDET-GEE-v2 on the other hand relies on two key parameters: i) a threshold which excluded steep pixels along the border of the flooded areas; ii) the number of time the borders of the flooded areas are recursively smoothed with a low-pass filter which reduces noisy and irregular topographical fluctuations (Cohen et al., 2022; Peter et al., 2020). Among the parameters values suggested by the authors, those found to perform the best across the two case study analyzed here are: 5°(about 9 %) for the threshold on the slope, and 5 for the number of iteration of the low-pass smoothing filter.

The average running times to generate results with FLEXTH on the same hardware described in Section 3 are: 30 seconds for the Brazos river (standard deviation: 15 seconds; n=10), and 45 minutes for the Tera-Orbigo-Esla confluence (standard deviation: 2 minutes; n=5). Average running times to generate the maps with FwDET on Google Earth Engine and different parameterizations are: 9 minutes for the Brazos river (standard deviation: 2 minutes; n=10), and 7.6 hours for the Tera-Orbigo-Esla confluence (standard deviation: 2.3 hours; n=5).

Figure 7 compares the water level and water depth estimated by the reference hydrodynamic model (first row), FLEXTH (second row) and FwDET (third row) for the Brazos river case. The result show how the FLEXTH estimates agree with the reference data. In particular, the fluctuations in the water level are smooth and realistic, as they do not present steep variations.

Analogously, Figure 8 displays the results in the lower part of the Tera-Orbigo-Esla confluence (about half of the overall flooded area). Also in this case, the water level and the water depth estimated by FLEXTH appear reasonable and in line with the reference hydrodynamic simulations.

Figure 9 quantifies the accuracy of the water depth estimates in both test sites. The result highlight the accurate estimates of FLEXTH, particularly in the Brazos river case, without displaying any substantial bias and showing reduced dispersion along the 1:1 line as compared to FwDET.

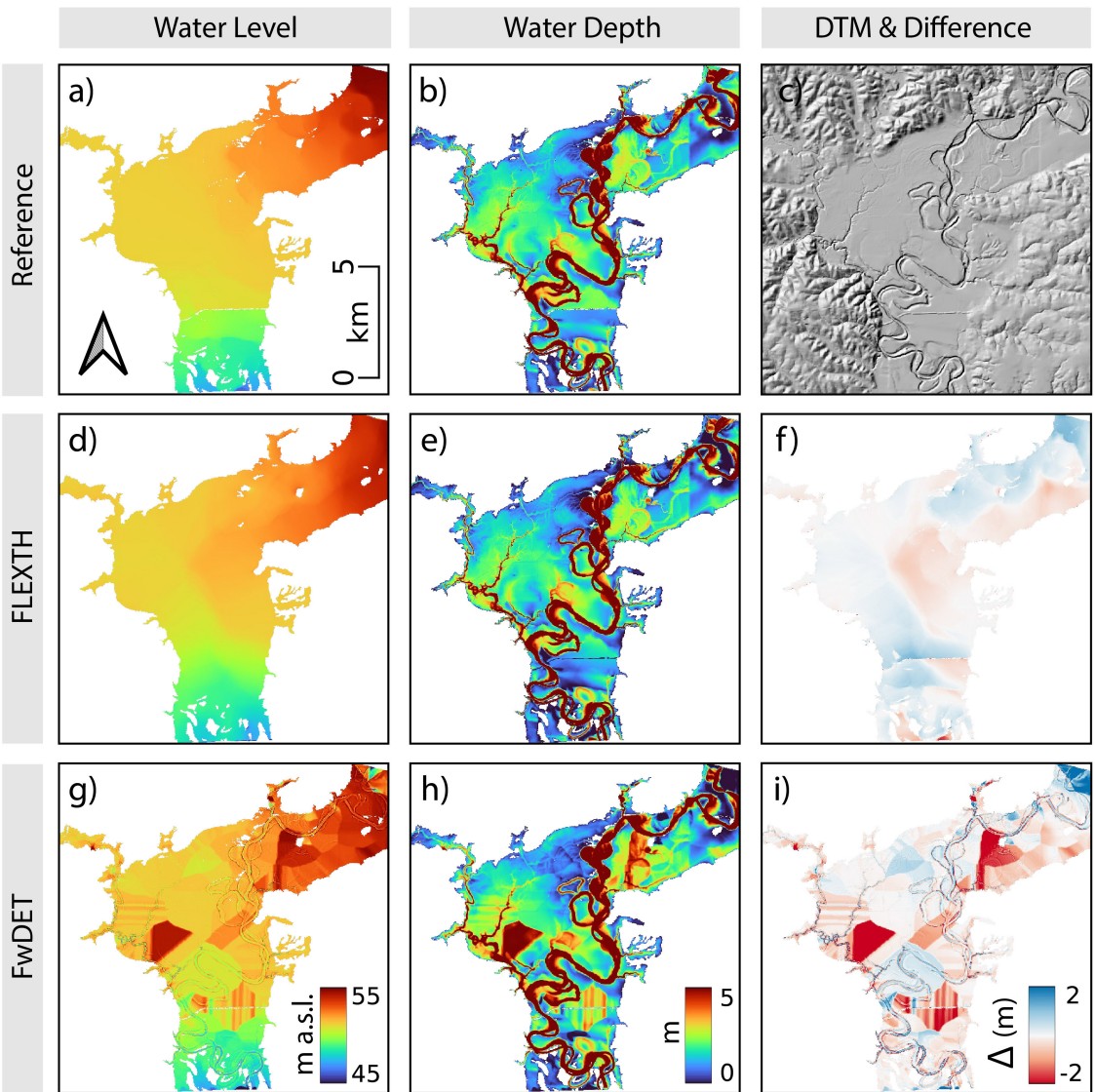

**Figure 7.** Brazos river: water level (first column) and water depth (second column) as estimated via the reference hydrodynamic simulation (first row, panels a and b), using FLEXTH (second row, panels d and e), and using FwDET (third row, panels g and h). Panel c shows the topography of the area (www.sciencebase.gov/catalog). Panels f and i display the deviation from the reference data (blue: underestimation; red: overestimation).

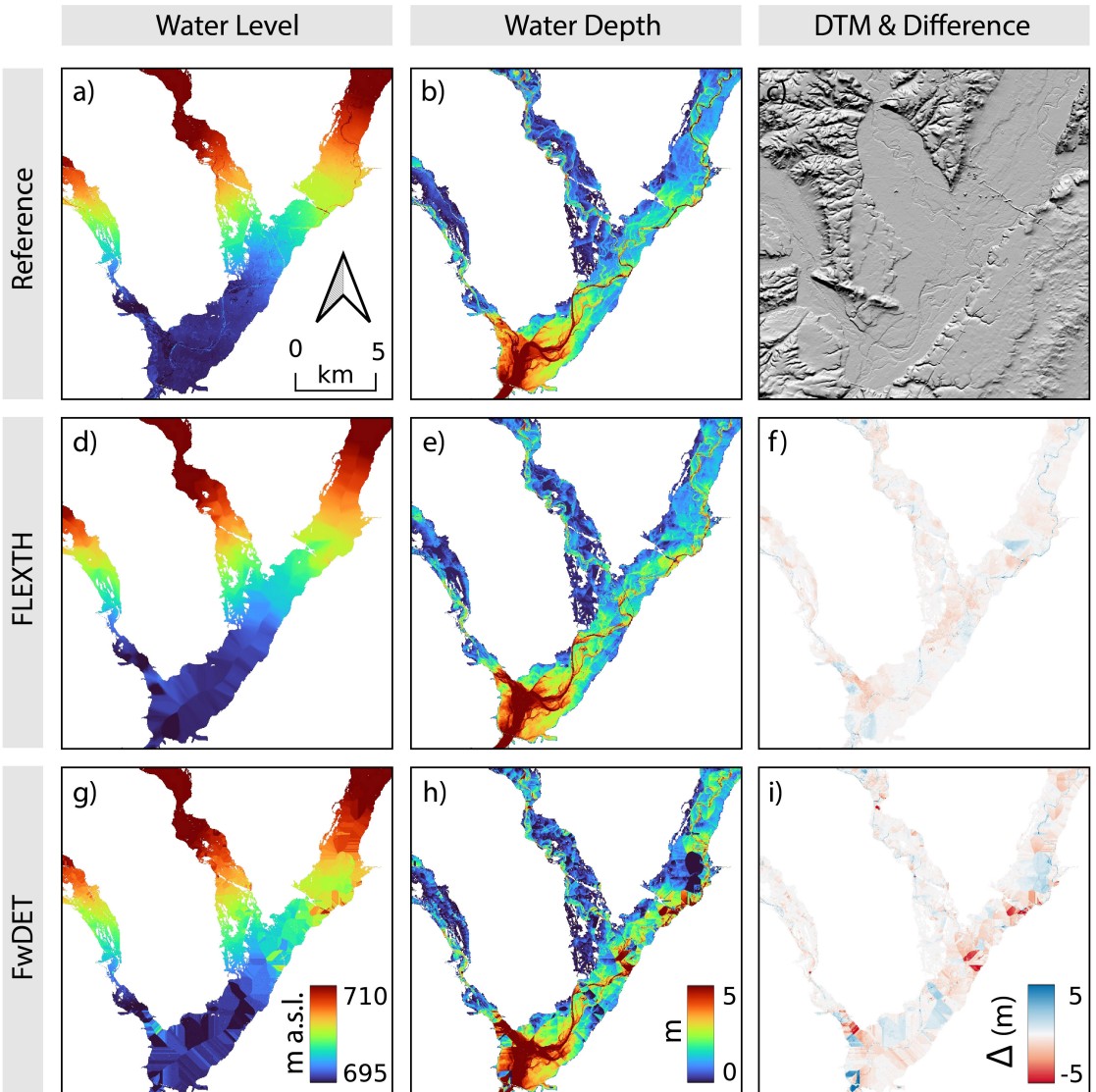

**Figure 8.** Tera-Orbigo-Esla confluence: water level (first column) and water depth (second column) as estimated via the reference hydrodynamic simulation (first row, panels a and b), using FLEXTH (second row, panels d and e), and using FwDET (third row, panels g and h). Panel c shows the topography of the area (https://centrodedescargas.cnig.es/CentroDescargas). Panels f and i display the deviation from the reference data (blue: underestimation; red: overestimation).

## 5 Evaluation of the flood propagation component of FLEXTH

A key feature of FLEXTH is its capability to propagate flood water across the areas where data is missing (see Section 2.3). This section systematically evaluates the flood propagation routine by quantifying how the final flood extent and water depth are impacted by the "invasiveness" of the no-data mask in the Brazos river case.

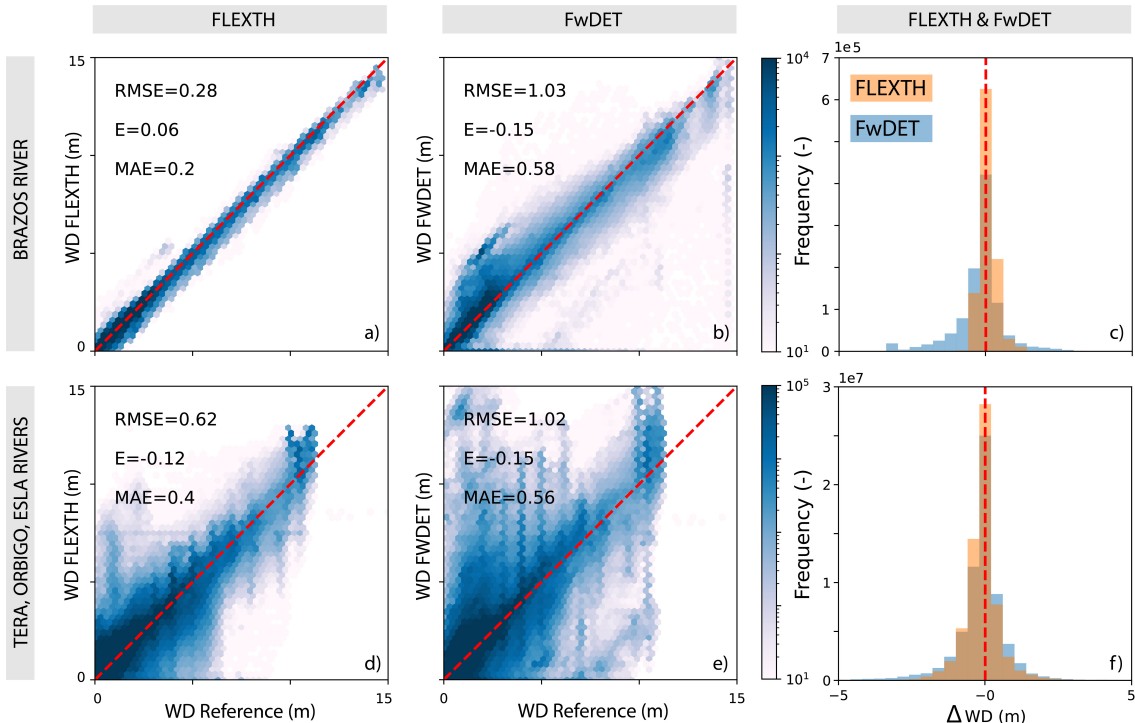

**Figure 9.** Density scatter plots comparing FLEXTH and FwDET water depth estimates against the reference data (i.e. hydrodynamic simulations) for the two test sites. $RMSE$, $E$ and $MAE$ denote the root-mean-square error, bias, and mean absolute error respectively. The histograms in panes c and f display the distribution of the differences $\Delta WD$ between the reference and the estimated water depths (positive values correspond to underestimations). Note that the performance metrics for the water level are identical as those for water depth because water depth and water level are linked via the same underlying topography.

For this purpose, 5 realizations of a set of 20 synthetic no-data masks are generated with increasing degree of masking. Each mask $M_i$, with $i = \{1, 2, ...20\}$ is obtained by recursively overlaying on the $M_{i-1}$ mask, a stochastic mask $m$. The mask $m$ is produced by first seeding 500 uniformly distributed centroids across the study area (i.e. $\{x_c, y_c\}_j$ , with $j = \{1, 2, ...500\}$,

$x_c \sim U(x_{min}, x_{max})$ and $y_c \sim U(y_{min}, y_{max})$, where $*_c$ are the locations of the centroids and $*_{min}, *_{max}$ delimit the spatial extent of the study area). Then, each centroid is associated with a stochastic radius $r_j$ sampled from an exponential distribution with average $R = 100\,m$. The area enclosed by each circumference of center $\{x_c, y_c\}_j$ and radius $r_j$ is masked. The initial baseline with no masking is denoted as $M_0$.

Some of the stochastic masks generated with this procedure are displayed in Figure 10, together with the corresponding

flood propagation and water depth estimated by FLEXTH, and the corresponding deviations from the reference dataset. For such simulation, FLEXTH receives as input: i) the DTM of the area; ii) the synthetic no-data masks; iii) the flood extents derived as described in Section 4 with no-data areas removed. The parametrization for the analysis is as in Table 1, except for

$A_{1/2}$ which is set to 10 km$^2$. The result show how both the flood extent and the estimated water depths reconstructed via the flood propagation component of FLEXTH correspond with the unmasked case, even for extremely extensive no-data coverage.

Figure 11 systematically evaluates the effect of increasing masking on: i) the absolute deviation from the reference flood extent; ii) the deviation of the water depth in correspondence to the masked areas. The figure shows that the initial unmasked flood extension could be reconstructed by the flood propagation routine of FLEXTH with a 10% accuracy up to a masking of 70%, and a mean absolute error of just 20 cm is committed even when half of the initial flood extent is missing data.

## 6    Discussion

Despite the complex flooding pattern in Pakistan, the water depth and water level estimates, together with the enhanced flood extent displayed in Figure 4 and 5, appear to be realistic, highlighting the potential of the procedure outlined in this study. The quantitative evaluation of the results confirms this first impression. Figure 5 shows that the Sentinel-1 derived flood delineation delivered by GFM already matches well the CEMS maps for all three AOIs (Table 2). Enhancing the flood delineation following the procedure described in this study, expands the flooded areas further, and reduces false negatives. Despite the increased

number of false positives due to the flood expansion, it should be noted that CEMS flood maps (used as "ground truth") are likely to suffer from underestimations, as they are derived based on the Airbus SPOT 6/7 optical sensor. In fact, similar to radar sensors, optical sensors may be inadequate to detect flood water under vegetation and in urban areas. Despite the efforts to identify and exclude areas where flood mapping cannot be performed reliably, CEMS products still have difficulty identifying all such areas, and some areas that are not excluded may actually be flooded. In contrast, the flood expansion procedure of

FLEXTH, because it has a physically-based component relying on ground topography, may be better suited for capturing floods in non-sensitivity areas.

Evaluating water depth estimates in flood maps is a challenging task: no direct or indirect measures of water level and/or water depth are generally recorded, especially over large areas and in underdeveloped regions. Nonetheless, the methodology proposed here to estimate and assess water depth highlights the value of ICESat-2 as a source of benchmark water depth

data. The 91-days revisit time of ICESat-2 reduces the chances of a temporal match between a flood peak and an altimetric acquisition. Furthermore, for the mid-latitudes (approximately 60S to 60N) laser beams operate off-nadir in order to increase spatial sampling. Considering the narrow diameter of laser pulses (about 17m), repeated cycles of the very same locations are practically impossible. Nonetheless, the current procedure, which considers two closely located acquisitions during the flood and non-flood periods, is reasonable for benchmarking purposes – at least on floodplains – because of the reduced topographic

gradients. The methodology developed to extract the reference altimetry with ICESat-2 described in section 3.1.2 appears robust for the Pakistan case (Figure 6c). However, other geographical conditions may require alternative procedures in order to remove buildings, vegetation and background noise from the raw ICESat-2 data (e.g. Neuenschwander and Pitts (2019); Li et al. (2020)), and further research is warranted to explore the full potential of ICESat-2 in remotely evaluating flood depth.

The limitations of FLEXTH are related to the accuracy and resolution both of the input flood delineation and of the un-

derlying DTM. Although flood depth at each pixel is estimated using the elevation of multiple locations along the contours

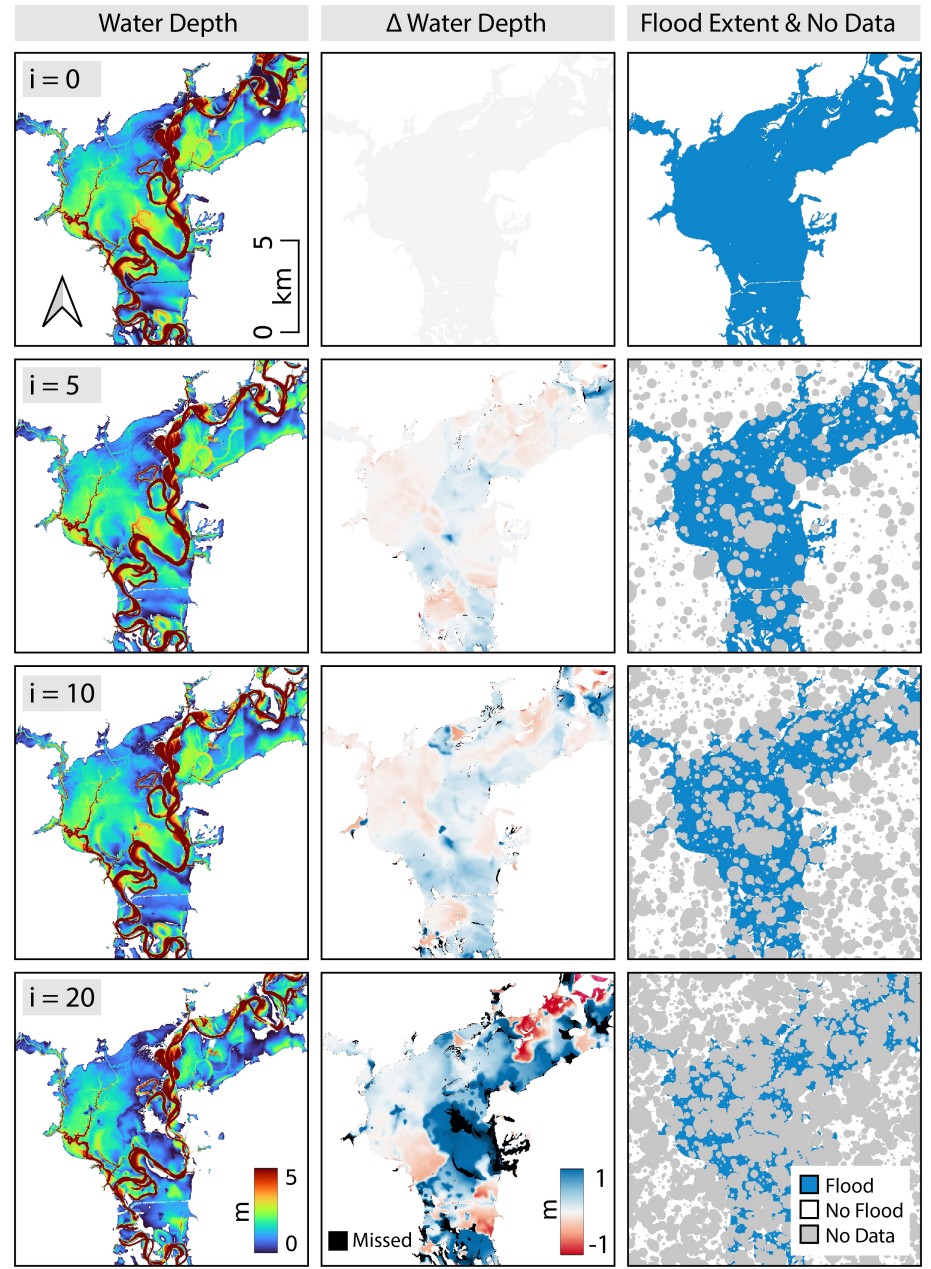

**Figure 10.** Water depth reconstructed via the flood propagation routine of FLEXTH and deviation with respect to the water depth estimated in the unmasked case for different levels of no-data masking (positive numbers correspond to underestimations).

of flooded areas in order to improve robustness, inaccuracies in the identification of the correct wet-dry divide may lead to erroneous water depth estimates, particularly in steep terrains. Large errors in water depth estimates are less likely in flat areas because of the reduced sensitivity of water level to potentially erroneous flood delineations. However, where topographical

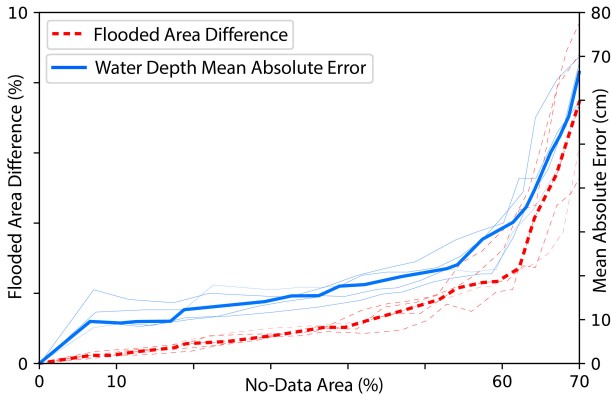

**Figure 11.** Red dashed line: percent of the reference flood extent which is not captured by the flood propagation component of FLEXTH for different degrees of masking. Blue solid lines: Mean absolute error of the water depth with respect to the reference case in the masked areas. The lines are the results of 5 realizations of the stochastic no-data mask (the thick lines are the medians.

gradients are reduced, even small differences in water level estimates can lead to floods propagating over very different extents. The DTM accuracy and resolution are also critical, as they are key in water level, and in turn, in water depth estimation. In fact, even assuming a perfect delineation of floodwaters, a coarse DTM in steep areas can strongly affect the resulting water levels.

It is also worth noting that the spatio-temporal merging of multiple flood delineation products can be inaccurate as it can generate singular features (e.g. sharp edges). This aspect is particularly relevant when the spatial scale of the flood is much larger than the imagery footprint, and in the case of low revisit times of the satellites.

The underestimated water depths shown in Figure 6b and e are likely to be related to underestimated flood extents and/or to the fact that the ICESat-2 acquisition was closer to the flood peak, compared with the Sentinel-1 imageries used by GFM to compute the initial flood extent. In fact, even using a larger percentile (say 0.75) of the distribution of ground elevation along the contour between flooded and non-flooded areas to estimate water levels, as per Method B (see Section 2.2), alleviates but does not solve the problem. This suggests that the underestimation is associated with the available input data, rather than a systematic negative bias of the methodology (as it is confirmed by Figure 9).

Figure 9 displays the good performance of the methodology in the more controlled scenarios provided by hydrodynamics simulations. The results are consistent, despite the reference benchmark dataset being derived from various geographical regions and obtained independently using different methodologies and resolutions. FLEXTH captures the variability of simulated water levels and water depths displaying a reduced bias (Figure 8 and 9). The relatively lower performances obtained in Spain is likely due to the different topographical information used for the current analysis (2 m DTM) as compared to that used by the Spanish authorities to perform the hydrodynamic simulations (i.e. 1 m DTM with extensive removal of bridges and other structures from the floodplain, see Section 4.1.2).

The reconstructed water level is fundamental to evaluate the plausibility of the results because it discounts for the additional variability introduced in the water depth by the underlying topography. Figure 7 and 8 show how FLEXTH properly simulates

a slow-varying water level across the study areas, particularly in the Brazos river case. The discontinuities in the Tera-Orbigo-Esla confluence can be attributed to the aspects noted in the previous paragraph, and can be reduced simply by increasing $N_{max}$ and/or reducing $\alpha$. On the other hand, FwDET is more vulnerable to local features in the DTM and/or in the flood delineation, resulting in unrealistic abrupt discontinuities in the estimated water depth. This aspect – noted by the authors in Cohen et al. (2019) – was attributed to the use of a single nearest-neighbor reference elevation during the interpolation phase. Despite the evolutions introduced in the latest release of FwDET increase the robustness of the results by filtering and smoothing the elevation of the outlines of the flooded areas (see Section 4), some artifacts still persist.

The flood expansion routine described in this study requires, as optional inputs, a no-data and a water body mask. While water depth estimation can still be performed in the absence of such inputs, it is worth noting that water level and water depth may be underestimated in areas featuring extensive interfaces between flood water and permanent waters, and/or where accurate flood mapping is impeded by dense vegetation or buildings. If the borders between flood water and no-sensitivity areas are not excluded, they will contribute to the estimation of water levels. In these settings, ground elevations used as a reference to compute water level will not correspond to the real wet-dry border (which in reality would be outside the flooded area and at higher elevation), therefore leading to underestimated water levels. Thus, especially in coastal flooding, it is paramount for the sea mask to cover (or properly match) the seaward outline of the flood delineation.

FLEXTH includes several user-defined parameters. Such parameters can enhance the power and the flexibility of the framework. The heuristically selected set of parameters in Table 1 led to adequate performances in multiple case studies and, overall, results are consistent across a wide range of parameter values. However, parameter selection may suffer from a degree of arbitrariness. For the water level and water depth estimation routine, the most critical parameters are found to be: $S_{max}$, $N_{max}$, $\alpha$ (and $P$ if Method B is used, see Section 2). The threshold for terrain slope, $S_{max}$ used to exclude steep pixels from being used as a reference for water level estimation, has to be a trade-off between too conservative (low) and too permissive (large). Low $S_{max}$ will reduce the number of pixels available for water level estimation, ultimately losing valuable altimetric information. On the other hand, a high $S_{max}$ may allow non-representative pixels to contribute for water level estimation. The selection of $S_{max}$ has also to account for the pixel size, as even a small threshold associated with a coarse DTM can lead to large vertical inaccuracies. The maximum number of pixels used to estimate water level at a target flooded location ($N_{max}$) is also relevant. Our tests showed that results are consistent, for $N_{max}$ in a range between 50 and 500 (larger values are suitable for higher resolutions), but at the upper end of the interval the role of $\alpha$ also becomes critical. Appropriate results have been found for $\alpha \in [1,3]$. Lower values of $\alpha$ use more information of the altimetry along a larger share of the flood borders and provide smoother water level estimates. However, they may also include distant borders that do not well represent water level at a given target location. On the contrary, by giving more weight to the nearest border pixels, high $\alpha$ can capture better the smaller-scale variability of the water level, although water level estimates can be more prone to local errors in the DTM and/or flood delineation. Larger $N_{max}$ and smaller $\alpha$ may be required to produce smoother and more realistic water level and water depth estimates when Method B is employed. The percentile $P$ of the distance-weighted distribution of reference pixels elevations used to estimate water level according to Method B (see Section 2.2), is also relevant. This parameter offers additional flexibility as it can be used to compensate for potential systematic biases of flood extent maps, or it can be employed to assess different

scenarios. Water level estimates using Method B can be more robust when flooded areas are surrounded by steep terrains, as water level is computed based on the percentiles of the altimetric distribution of the wet-dry contour. However, under some topographical conditions, using large percentiles (say above 75 %) may lead to erroneous and exaggerated flood propagation, triggered by overestimated initial water levels. On the other hand, our tests showed limited sensitivity of the results on the shapes and sizes of the filtering kernels, as well as on the type of connectivity used for the spatial analyses (see Section 2.2 and

2.3).

The flood propagation routine itself should also be critically assessed. Despite following simple physically based principles, it extrapolates information where no flood mapping is available. Especially in the presence of regional topographical gradients and extensive excluded areas, floods can potentially propagate across large areas and up to large distances. Improved flood mapping, DTMs, and limiting the extent of no-data regions, will constrain flood propagation and limit potential misbehaviours.

Nonetheless, the two parameters controlling flood propagation offer adequate flexibility to control flood propagation under different circumstances, and can be tweaked as desired. The systematic analysis of the impacts of no-data on the performance of FLEXTH (Section 5) clearly shows the added value of the methodology. In fact, the original flood extent and water depth in the use case is adequately reproduced even for extremely invasive no-data areas.

Additional analyses are envisaged to further explore the role of model parameters. However, it has to be stressed that ideal

parameter values may depend on local conditions, and in particular on the accuracy and resolution of both the available flood delineation map and of the underlying DTM. For this reason, a "one-fits-all" approach in selecting the optimal parameter might be unrealistic.

Finally, it is worth stressing that the approach can be applied to any flood delineation (not necessarily satellite-based) and to any flooding mechanisms, including floods in coastal settings. In fact, coastal areas can present favorable conditions for

the application of the methodology. In general, coasts do not display topographical gradient which may challenge the flood propagation routine as the regional ground elevation tend to increase landward (constraining flood expansion). Furthermore, in coastal areas, FLEXTH has to ingest a reduced variability of initial water levels as compared to riverine environments as surge run-up can range up to about 10 m (Fritz and Okal, 2008; Fritz et al., 2009, 2010; Liu et al., 2005). On the other hand, water levels along a flooding river can span hundreds of meters, depending on the regional slope and the scale of the event (e.g. over

100 m in the Pakistan case study).

## 7   Conclusions

The study presented a robust methodology – named FLEXTH – to estimate flood depth and to improve flood delineation based on inundation maps, readily available DTMs, and open source tools. The procedure requires minimum supervision and can run over extremely large areas in a reasonable amount of time.

The workflow starts by identifying suitable wet-dry boundaries from flood maps, which can be derived by any means (e.g. by satellites or aerial sensors, or ground surveys). By combining flood boundaries that are altimetrically informative with digital models of the land surface, the procedure extracts the ground elevation along the borders of connected flooded areas, regardless

of the complexity of the flood outlines and the surrounding topography. At each flooded location, water levels are computed based on the distance-weighted reference elevations of multiple points along the wet-dry border. Water is then propagated in contiguous low-lying no-data areas via a novel procedure which results in new dry-wet borders informed by the topography. Hydraulic connectivity is guaranteed by a recursive propagation algorithm, which additionally enforces a routine to realistically decrease water levels as propagation advances. Finally, flood water depths are obtained subtracting the underlying topography from the water surface elevation.

Flood extents, computed for the 2022 flood disaster that hit the Indus valley in Pakistan, are shown to match adequately the available benchmark data. Furthermore, the reconstructed water depths match the reference data obtained with a procedure employing the data acquired by the space-borne lidar onboard the ICESat-2 mission. Additional tests against hydrodynamic simulations in different settings provided good result, particularly highlighting the capabilities of the flood propagation component of FLEXTH, which can reconstruct flood extent and water depth even when extensive areas lack of thematic information.

These results are an encouraging starting point for a systematic application of the framework over large-sized data sources of flood extent maps, such as those from the recently released Global Flood Monitoring of the Copernicus Emergency Management Service. Future development of the methodology will focus on systematically optimizing the parameters of the algorithm under different geographical conditions. Fine-tuning the architecture of the algorithm is also envisaged for future releases, possibly targeting the computational speed of the flood propagation component. It is finally worth mentioning that FLEXTH can be effectively applied also to coastal floodings, provided that a mask is assigned covering the sea surface.

Overall, the presented methods can assist emergency response planning, flood impact assessment, and contribute to reduce the disastrous consequences of floods worldwide, especially in vulnerable regions where flood risk is exacerbated by a changing climate.

*Code availability.* FLEXTH is available as a Python script at https://code.europa.eu/floods/floods-river/flexth. An additional script named "DTM_2_floodmap.py" is also provided to easily resample/reproject and align input data into the same grid and projected reference system as the flood delineating raster.

*Data availability.* All data used for this study are publicly available. The results of the hydrodynamic simulations in the Brazos river are available on request upon A. Betterle.

*Author contributions.* Conceptualization: AB; Data curation: AB; Formal analysis: AB; Investigation: AB; Methodology: AB; Project administration: PS; Resources: PS; Software: AB; Supervision: PS; Validation: AB; Visualization: AB; Writing - original draft preparation: AB; Writing: review and editing: AB, PS

*Competing interests.* The authors declare no conflict of interests.

*Acknowledgements.* We thank S. Cohen and an anonymous reviewer for the valuable criticism during the reviewing process of this study. We also wish to thank J.-F. Pekel, P. Ceccato and N. McCormick for providing comments on an early version of the manuscript, J. Casado Rodriguez for the help with the data of the Spanish test case, and A. Haag for an early discussion on satellite altimetric data.

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
