# Peer review of "Water depth estimate and flood extent enhancement for satellite-based inundation maps"

_Natural Hazards and Earth System Sciences, 2024_

## Author Comment (AC1)

**Reviewer 1**

Dear authors,

I have read your manuscript entitled "Water depth estimate and flood extent enhancement for satellite-based inundation maps" with great interest.The manuscript's focus on the development of the FLEXTH algorithm to address the limitations of existing flood mapping methodologies is commendable. The algorithm's utilization of topographic information for enhancing flood delineation and providing estimates of water level and depth across entire flood extents represents an advancement in the field. However, the discussion on the algorithm's key features, such as accuracy, limited supervision requirements, and computational efficiency, lacks credibility due to the weakness or absence of supporting evidence. As a result, its potential applicability in large-scale flood assessments is called into question.

The introduction section offers a comprehensive review of methodologies for estimating flooded area and water depth, highlighting both the limitations and advancements in current approaches. This provides valuable insights into the current state of the field. However, apparent lack of awareness regarding some of the latest developments in the field casts doubt on the claims of novelty surrounding the method presented in the manuscript.

Overall, the paper is well-written, logically structured, and most of the figures are appropriate. The introduction of the FLEXTH algorithm represents a notable contribution to the field, with the potential to enhance flood assessment and disaster response strategies. The open access Python script is also a welcome addition to facilitate further research and collaboration within the scientific community.

Detailed below are some specific comments. I strongly suggest a significant revision of this manuscript to address these issues thoroughly.

We express our gratitude to the Reviewer for the interest in our study and for investing time in conducting a thorough assessment. We value the recognition of our methods as a significant contribution to the field, as well as the positive assessment of the manuscript's clarity and logical structure. Additionally, we are pleased that our commitment to open science, demonstrated by making the code freely accessible, has been acknowledged.

In revising the manuscript, we are specifically addressing the issues concerning the "accuracy, limited supervision requirements, and computational efficiency, lacks credibility due to the weakness or absence of supporting evidence". In particular, in order to address the remark about the limited validation of the method, and the effect of the no-data areas on the flood propagation routine, we will include two new section to the manuscript. The first providing a comprehensive comparison of the methodology against what is probably the state of the art in the field. The section will include ad-hoc hydrodynamic simulations (which will be made openly available to the scientific public given the notorious lack of openly available dataset for validation purposes). The

second new section will systematically assess the effect of the no-data areas on the performances of the algorithm, in particular on the flood propagation routine.

Finally we will revise the references as suggested and modified the two figures where improvements were necessary.

We hope the revised manuscript will meet the expectation of the Reviewer, as we are planning to incorporate most of his/her request.

Below each comment is addressed in details.

Detailed comments:

1. Line 4's mention of "billions of euros" seems Europe-centric, overlooking the global nature of flooding, which disproportionately impacts lower socio-economic regions over developed countries. It would be beneficial to provide a more inclusive perspective on the economic impacts of flooding, considering the varying economic contexts and vulnerabilities worldwide. Additionally, emphasizing the socioeconomic disparities exacerbated by flooding in vulnerable regions can underscore the urgency of addressing this global challenge.

   We acknowledge the remark. We will remove the reference to euro and added a sentence to stress that floods have disproportionate impacts in less developed areas.

2. The term "Exclusion Mask" introduced in Line 51 and used throughout the manuscript pertains specifically to the Global Flood Monitoring (GFM) product. However, a more suitable term might be "No Data Areas," as this is commonly encountered in satellite-derived flood extents regardless of the satellite or product used. While it's assumed that no data areas can be treated similarly to the exclusion mask mentioned here, this assumption should be explicitly addressed.

   Agreed, we will exchange the "Exclusion Mask" notation.

3. The assertion in line 75 regarding the unproven applicability of water depth estimation approaches for large-scale assessments is inaccurate. For instance, Teng et al. (2022) (https://doi.org/10.1029/2022WR032031) have conducted a comprehensive comparison of various methods and conclusively demonstrated their effectiveness for large-scale assessments. Peter et al. (2022) (https://doi.org/1 0.1109/LGRS.2020.3031190) have also implemented FwDET into Google Earth Engine for rapid and large-scale flood analysis. It

is imperative to acknowledge and incorporate these findings to ensure the accuracy and completeness of discussion on this matter.

Thanks for pointing that out. These studies will be acknowledged. Furthermore, the revised manuscript will features new sections including a comparison between FLEXTH and the Google earth engine implementation of FwDET (the one of Peter et al., 2022 mentioned by the reviewer in the comment).

4. In lines 87-89, the claim regarding computational efficiency uses the term "for areas of up to tens of thousands of square kilometres", which lacks rigor and specificity, particularly in terms of resolution. It is recommended to provide the number of grid cells or include the specific resolution considered to accurately assess its significance in the context of flood mapping. In addition, you are making a claim about the computational advantages of the proposed method without providing solid evidence or comparisons with other existing approaches in terms of run speed. It's essential to provide empirical data or benchmarks to support this claim and accurately assess the computational efficiency of the proposed method relative to other methods in the field.

We agree with the comment (scale without resolution is not very informative). We will specify the issue. The first new section will includes new simulation and comparison metrics with the widely used FWDet V2.0 (Cohen et al.,2019; Peter et al.,2022) specifically addressing these aspects.

5. I find it difficult to follow Figure 1 in its current horizontal layout. Consider changing it to a vertical layout and using standard flow chart shapes to improve clarity and ease of understanding.

We will try to improve the figure following the suggestion.

6. Section 2.1 is titled "Input and Output Products" but does not mention output at all. Consider revising the title to accurately reflect the content or include information about output products in the section.

Agreed, thanks for pointing that out.

7. In the paragraph starting from Line 156, Method A is effectively using Inverse Distance Weighting (IDW), while Method B seems to be a crude percentile-based averaging algorithm. It would be beneficial to consider other interpolation methods such as Spline,

Kriging, or more advanced machine learning methods. These alternative approaches may offer advantages in terms of accuracy, robustness, and flexibility, especially in handling complex spatial relationships and varying data distributions. Therefore, exploring and comparing these different interpolation techniques could provide a more comprehensive understanding of the flood water level along the dry-wet borders and potentially improve the accuracy of the results.

In method B the distribution depends on the distance of each border pixel from each target location inside the flooded area where water levels are computed. In this way the impact of each reference elevation in the distribution is weighted based on its distance and the ensuing distribution is substantially different from the mere un-weighted version.

About the interpolation methods we appreciate the remark. In fact, other interpolating methods have been considered (including some of those mentioned by the Reviewer). However, they require more tweaking, and, in our tests, they did not evidence any substantial advantage, particularly considering that they are not so robust for systematic, unsupervised and large-scale applications (as they are more sophisticated). We also would like to point out that standard Kriging methods (e.g. simple and ordinary Kriging) won't be ideal for applications where a regional topographic gradient is present (as it would be the case for large-scale applications). In these settings, more refined versions of Kriging would be necessary, where the regional trends can be properly accounted for (e.g. universal Kriging or Kriging with drift). Such methods would increase the degree of complexity and might cast doubts on the overall robustness of the methodology.

8. Figure 2(especially C) requires additional clarification to enhance its interpretability in its current form. Providing detailed annotations, labels, and a clear legend could help elucidate the information presented and make the figure more intuitive for readers to understand. Additionally, including a brief description of the data represented in Figure 2C within the main text could provide context and aid in interpretation.

Thanks for rising this issue. We will revise the figure following the suggestion.

9. The approach introduced in Section 2.3 is novel. It delineates new dry-wet borders informed by DTM in excluded or no data areas. This method, while simple, represents a step forward and deserves emphasis as the main novelty of this manuscript. However, the manuscript does not sufficiently demonstrate the effectiveness of the flood propagation routine. To address this, it is recommended to block out areas of flood extent and propagate flood into those areas as if they were excluded, then compare the results with the actual border. This step is critical to substantiate the effectiveness of this novel method.

Thanks for appreciating the novelty of the methodology. This aspect will be highlighted.

In order to address the doubt of the Reviewer concerning the propagation routine (also shared with the second Reviewer), we will include a new section to the revised manuscript, which will specifically address the effect of masking on the performances of the algorithm. We are confident the reviewers will appreciate the new systematic analysis.

10. One significant critique of this study is its reliance on a single case study: the Pakistan 2022 case study. This limited scope is insufficient, particularly considering the lack of easily accessible validation data for this specific case study. Moreover, it hinders the ability to compare the method's accuracy and computational efficiency with other existing approaches. To address this limitation, it is recommended to include additional case studies using published datasets. This would allow for a more convincing demonstration of the advantages of the proposed method.

Unfortunately freely available data sources, especially when it comes to water depth, are usually not openly available. See for example the review paper mentioned in comment #3 (Teng et al. (2022)). That study does not provide the modelled water depth as the access to such data is restricted.

Acknowledging the lack of validation data as a critical aspect, the revised manuscript will include a new section dedicated to assess FLEXTH against hydrodynamic simulations in 2 geographically different locations. Although hydrodynamic simulations do not necessary reproduce real-world cases, they provide realistic physically-based scenarios useful for validation purposes and they readily provide flood extents and water depths, circumventing the limitation of remote-sensing methodologies.

11. Lines 234-235, how does this run speed compare to other existing approaches? You are claiming computational advantages without solid evidence.

We are planning to provide further evidences to support our claims in the revised version.

12. Line 243: Please clarify the meaning of CEMS. Please spell out acronyms before their first reference. Similarly, for EMSR629, FABDAM, and other acronyms, provide their full expansion before their initial mention.

Thanks for the remark. We will fix the issue.

13. You are using one satellite product to validate another satellite product of flood extent, which warrants clarification. Please explicitly state the advantage of CEMS over GFM flood extents for validation purpose.

This is true but it is the only possible way unless ground-based flood delineations are available (unrealistic) or if aerial-based flood delineation are provided (rare and with difficult data access). An alternative would be to use numerical models (which we will employ in the revised manuscript, see the response to comment #10). In the current text it is specified that the CEMS flood delineation are semiautomatic with expert supervision/refinement. GFM on the other hand is a fully automatic and unsupervised system (with the consequent limitations). Following the Reviewer's comment we will try to stress the aspect better in the revised manuscript.

14. In addition to the metrics listed in Table 2, it would be recommended to include F-stat as an additional accuracy metric.

Agreed, we will included the metric suggested by the Reviewer.

15. Using ICESat-2 altimetry data as truth to validate water depth estimates could be problematic due to mismatching of footprint and timing, as you have discussed in Section 4. To mitigate this concern, it would be advisable to incorporate additional case studies with more suitable validation data, as mentioned in the comments above.

The use of ICESat-2 to validate flood depth estimates is per se a novelty in the field which was probably not stressed sufficiently (as noted by the second Reviewer). In fact, we are not aware of any published study that uses similar methods. Errors due to the spatial mismatch are minimal (as discussed in the manuscript). Errors due to the temporal mismatch between GFM flood maps and ICESat-2 acquisitions are also mentioned and, as correctly pointed out by the Reviewer. However, we would like to underline that similar problem would occur regardless the source of the satellite products used for validation. This happens, for example, when validating flood extent products with other products having higher mapping capabilities.

To address this issue the revised manuscript will include a new section dedicated to validation which will employ hydrodynamic simulations in 2 additional case studies.

16. Line 322 acknowledges the critical importance of DTM accuracy and resolution. However, it raises the question of why the study does not utilize high-resolution and high-accuracy DEM data, which are available in many regions globally. If data accessibility is an issue,

it would be more beneficial to include additional case studies that utilize such data to enhance the robustness and applicability of the findings.

We believe that showing the performances of the methodology even without using site-specific high resolution topographical data is an additional proof for the suitability of the methods for large scale applications. As suggested for similar methods (see Cohen et al., Remote sensing, 2022) higher quality DTM will systematically improve the performances of the methodology.
The new test cases that will be included in the revised manuscript will use DTM with different resolution, thus helping clarifying this aspect.

17. Thank you for providing the source code. After reviewing the code, I was unable to identify memory control or chunking algorithms that would support your claims of computational efficiency for large-scale studies. Could you please clarify this aspect?

The revised manuscript will address this aspect. Thanks for pointing that out.

18. Validation data have not been provided along with the source code.

All the validation data are freely accessible online on GFM and ICESat-2 portals( https://portal.gfm.eodc.eu/login?redirect=%5Bobject%20Object%5D ; https://openaltimetry.earthdatacloud.nasa.gov/data/index.html). FABDEM is also publicly accessible (https://data.bris.ac.uk/data/dataset/s5hqmjcdj8yo2ibzi9b4ew3sn). We don't see the point to include in a dedicated repository a duplicate of these bulky data. However, we will specify in the revised manuscript where the data can be retrieved. What we are planning to make available instead is the result the ad-hoc hydrodynamic simulation that we performed to assess the performances of the algorithm. We believe that the latter data is more critical given the notorious lack of validation/testing datasets for flood-related studies of this type.

---

## Author Comment (AC2)

**Reviewer 2**

The manuscript presents a new algorithm for improving and enhancing remote sensing-derived flood inundation maps. The algorithm was developed to address limitations in the Copernicus Global Flood Monitoring (GFM) system but can be used for other applications. The algorithm fills gaps in the flood maps by expanding the flooded domain into the GFM 'Exclusion mask' area and calculates water depth. The approach builds nicely on recent efforts in this field and presents a considerable advancement. The manuscript is very well written and the algorithm evaluation analysis is well-reasoned and presented. I have two main concerns: (1) the evaluation benchmark is a flood map (and water depth) derived from different satellite sensors which likely suffers from somewhat similar (or other form of) biases as the SAR map; (2) the algorithm description is not as clear as it should be. The authors acknowledge the issues with the benchmark dataset used for evaluation and frame the results in this context. They also rightly assert that robust data for large-scale flooding is scarce. One remedy is to enhance the AOI-based analysis and add locations in which there is higher confidence in the benchmark data.

We thank the Reviewer for appreciating the study and to stress some limitations which will be addressed in the revised version. In particular, an additional section has been added specifically to address point (1) above (such aspect was also stressed by Reviewer 1). In this new section, the results from virtual flood scenari obtained via hydrodynamic simulations have been used as a reference to benchmark the estimates provided by FLEXTH. Hydrodynamic simulation in fact are suitable for the purpose because: i) provide a clear delineation of flooded areas (which is not always possible based on satellite products because of no sensitivity and/or water-look-alike conditions); ii) provide estimates of water depth and water level which can be used for benchmarking. We furthermore improved the description of the algorithm (point (2) above), particularly flowing the remarks provided by the Reviewer in the following comments.

An additional new section will be added focusing on the effect of the extension of the exclusion mask on the final results. In fact both reviewers showed interest/criticism for this aspect.

A series of new figures will support the new sections.

Specific issues:

Line 118: clarify 'last flooded pixel'

The sentence will rephrased and clarified.

Line 127: what do you mean by '"morphological" closing'?

The technical term will be specified in the revised text.

Section 2.2 is not clear enough.

We will attempt to improve the section, although it is not specified what was unclear.

The small paragraph starting in line 132 is not very clear.

The paragraph will be rephrased.

Figure 2 can be better explained and referenced in the text.

The description of the Figure will be extended and an additional reference provided.

Line 154: not sure what you mean by '4-connectivity'

The meaning of the term will be clarified.

Figure 3: clarify if these are purely synthetic (1D) results (i.e. not using the full algorithm)

Agreed, the aspect will be specified.

I don't recall you explicitly explaining when the recursion ends (I assume when water cannot be further propagated). It is also not entirely clear the 'seeding' of the reduction - is every pixel at the edge of the flood a seed?

Thanks for pointing out the aspect. The revised version of the text will better specify the methodology.

Table 2 and text: consider reporting the overall improvement in terms of %.

The improvement will be reported in the main text.

Line 235: how many pixels and what is the resolution used?

The information will be included.

Line 272: I don't understand what 'i=1,2' stands for.

We will clarify the confusion in the revised text.

Line 289: consider removing 'To conclude'

Agreed.

Figure 6: what is RMSE represent - error compared to what?

Thanks for pointing that out. The comparison is with respect to the ideal case (Delta = 0). This will be specified in the revised manuscript.

The use of ICESat-2 altimetry data to evaluate the water depth prediction is quite novel (to my knowledge) and could be of great interest. The limitations in the data acquisition and processing is well described but additional emphasis needs to be made on the limitations and unknowns associated with your approach - highlighting the need for additional research focused on this approach.

Thanks for appreciating the novelty of the methodology. We will address the Reviewer's comment by providing some additional considerations in the revised text. For example it can be mentioned that the ICESat raw data requires some filtering to remove the effect of objects on the ground. This can have some impacts.

Line 394: 'hydraulic connectivity' is a good term to use in this context - consider adding it to the algorithm description.

We agree, thanks for pointing that out!

Line 403: 'FLEXTH can [also] be effectively applied to...'

Agreed.

The algorithm applies the expansion procedure to masked areas which you found to be a potential limitation. Did you look into running it without this limit?

The effect of the exclusion mask on the performances of FLEXTH and particularly on its propagation routine will be systematically assess in a dedicated new section of the revised manuscript. We deem this as an interesting aspect which was also stressed by the first reviewer. We are confident that the reviewers will apprecia

---

## Author Response (AR1)

**Reviewer 1**

Dear authors,

I have read your manuscript entitled "Water depth estimate and flood extent enhancement for satellite-based inundation maps" with great interest.The manuscript's focus on the development of the FLEXTH algorithm to address the limitations of existing flood mapping methodologies is commendable. The algorithm's utilization of topographic information for enhancing flood delineation and providing estimates of water level and depth across entire flood extents represents an advancement in the field. However, the discussion on the algorithm's key features, such as accuracy, limited supervision requirements, and computational efficiency, lacks credibility due to the weakness or absence of supporting evidence. As a result, its potential applicability in large-scale flood assessments is called into question.

The introduction section offers a comprehensive review of methodologies for estimating flooded area and water depth, highlighting both the limitations and advancements in current approaches. This provides valuable insights into the current state of the field. However, apparent lack of awareness regarding some of the latest developments in the field casts doubt on the claims of novelty surrounding the method presented in the manuscript.

Overall, the paper is well-written, logically structured, and most of the figures are appropriate. The introduction of the FLEXTH algorithm represents a notable contribution to the field, with the potential to enhance flood assessment and disaster response strategies. The open access Python script is also a welcome addition to facilitate further research and collaboration within the scientific community.

Detailed below are some specific comments. I strongly suggest a significant revision of this manuscript to address these issues thoroughly.

We express our gratitude to the Reviewer for the interest in our study and for investing time in conducting a thorough assessment. We value the recognition of our methods as a significant contribution to the field, as well as the positive assessment of the manuscript's clarity and logical structure. Additionally, we are pleased that our commitment to open science, demonstrated by making the code freely accessible, has been acknowledged.

In revising the manuscript, we specifically addressed the issues concerning the "accuracy, limited supervision requirements, and computational efficiency, lacks credibility due to the weakness or absence of supporting evidence". In particular, in order to address the remark about the limited validation of the method, and the effect of the no-data areas on the flood propagation routine.

To address the reviewer's comments, two new sections have been included to the revised manuscript, including 5 new figures (See Sections 4 and 5 and Figures 7-11). The first section provides a comprehensive comparison of the methodology against hydrodynamic simulation and

against the Google Earth Engine implementation of FwDET v2, which is probably the state of the art in the field.

The second new section (Section 5) systematically assess the effect of the no-data areas on the performances of the algorithm, particularly focusing on the flood propagation routine.

Finally we have revised the references as suggested.

We hope the revised manuscript will meet the expectation of the Reviewer, as it incorporate most of his/her request.

Below each comment is addressed in details.

Detailed comments:

1. Line 4's mention of "billions of euros" seems Europe-centric, overlooking the global nature of flooding, which disproportionately impacts lower socio-economic regions over developed countries. It would be beneficial to provide a more inclusive perspective on the economic impacts of flooding, considering the varying economic contexts and vulnerabilities worldwide. Additionally, emphasizing the socioeconomic disparities exacerbated by flooding in vulnerable regions can underscore the urgency of addressing this global challenge.

   We acknowledge the remark. We have rephrased the sentence removing the reference to euro and stressing the disproportionate impact of floods on less developed regions.

2. The term "Exclusion Mask" introduced in Line 51 and used throughout the manuscript pertains specifically to the Global Flood Monitoring (GFM) product. However, a more suitable term might be "No Data Areas," as this is commonly encountered in satellite-derived flood extents regardless of the satellite or product used. While it's assumed that no data areas can be treated similarly to the exclusion mask mentioned here, this assumption should be explicitly addressed.

   Agreed, the term "exclusion" has been substituted with no-data throughout the manuscript.

3. The assertion in line 75 regarding the unproven applicability of water depth estimation approaches for large-scale assessments is inaccurate. For instance, Teng et al. (2022) (https://doi.org/10.1029/2022WR032031) have conducted a comprehensive comparison of various methods and conclusively demonstrated their effectiveness for large-scale assessments. Peter et al. (2022) (https://doi.org/1 0.1109/LGRS.2020.3031190) have also

implemented FwDET into Google Earth Engine for rapid and large-scale flood analysis. It is imperative to acknowledge and incorporate these findings to ensure the accuracy and completeness of discussion on this matter.

Thanks for pointing that out. The studies have been acknowledged and added to the references. Furthermore, the revised manuscript features the new Section 4 providing a detailed comparison between FLEXTH and the Google Earth Engine implementation of FwDET (the one of Peter et al., 2022 mentioned by the reviewer in the comment).

4. In lines 87-89, the claim regarding computational efficiency uses the term "for areas of up to tens of thousands of square kilometres", which lacks rigor and specificity, particularly in terms of resolution. It is recommended to provide the number of grid cells or include the specific resolution considered to accurately assess its significance in the context of flood mapping. In addition, you are making a claim about the computational advantages of the proposed method without providing solid evidence or comparisons with other existing approaches in terms of run speed. It's essential to provide empirical data or benchmarks to support this claim and accurately assess the computational efficiency of the proposed method relative to other methods in the field.

We agree with the comment (scale without resolution is not very informative). The first new section includes new simulation and comparison metrics with the widely used FWDet V2.0 (Cohen et al.,2019; Peter et al.,2022) specifically addressing these aspects (see sections 4.1.1, 4.1.2, 4.2; for example lines 219, 323, 332, 340-344).

5. I find it difficult to follow Figure 1 in its current horizontal layout. Consider changing it to a vertical layout and using standard flow chart shapes to improve clarity and ease of understanding.

Thanks for the suggestion, but we prefer to keep the figure compact like it is. However we solved a rendering issue with the underlying pdf version that might improve the readability of the figure (especially if it is printed).

6. Section 2.1 is titled "Input and Output Products" but does not mention output at all. Consider revising the title to accurately reflect the content or include information about output products in the section.

Good point! A mention to the output products is now available in the end of the section.

7. In the paragraph starting from Line 156, Method A is effectively using Inverse Distance Weighting (IDW), while Method B seems to be a crude percentile-based averaging algorithm. It would be beneficial to consider other interpolation methods such as Spline, Kriging, or more advanced machine learning methods. These alternative approaches may offer advantages in terms of accuracy, robustness, and flexibility, especially in handling complex spatial relationships and varying data distributions. Therefore, exploring and comparing these different interpolation techniques could provide a more comprehensive understanding of the flood water level along the dry-wet borders and potentially improve the accuracy of the results.

In method B the distribution depends on the distance of each border pixel from each target location inside the flooded area where water levels are computed. In this way the impact of each reference elevation in the distribution is weighted based on its distance and the ensuing distribution is substantially different from the mere un-weighted version.

About the interpolation methods we appreciate the remark. In fact, other interpolating methods have been considered (including some of those mentioned by the Reviewer). However, they require more tweaking, and, in our tests, they did not evidence any substantial advantage, particularly considering that they are not so robust for systematic, unsupervised and large-scale applications (as they are more sophisticated). We also would like to point out that standard Kriging methods (e.g. simple and ordinary Kriging) won't be ideal for applications where a regional topographic gradient is present (as it would be the case for large-scale applications). In these settings, more refined versions of Kriging would be necessary, where the regional trends can be properly accounted for (e.g. universal Kriging or Kriging with drift). Such methods would increase the degree of complexity and might cast doubts on the overall robustness of the methodology for extensive unsupervised applications.

8. Figure 2(especially C)requires additional clarification to enhance its interpretability in its current form. Providing detailed annotations, labels, and a clear legend could help elucidate the information presented and make the figure more intuitive for readers to understand. Additionally, including a brief description of the data represented in Figure 2C within the main text could provide context and aid in interpretation.

Thanks for rising this issue. Although we find Figure 2 already quite cluttered and we prefer not to add further information directly to it, we added extensive references to the steps described in the Figure throughout section 2.2 and 2.3 (see the revised version of the manuscript). In fact, in our opinion, the two sections describe already in detail all steps depicted in Figure 2. Each panel of the figure is now explicitly refereed to throughout the main text.

9.  The approach introduced in Section 2.3 is novel. It delineates new dry-wet borders informed by DTM in excluded or no data areas. This method, while simple, represents a step forward and deserves emphasis as the main novelty of this manuscript. However, the manuscript does not sufficiently demonstrate the effectiveness of the flood propagation routine. To address this, it is recommended to block out areas of flood extent and propagate flood into those areas as if they were excluded, then compare the results with the actual border. This step is critical to substantiate the effectiveness of this novel method.

    Thanks for appreciating the novelty of the methodology. This aspect has been highlighted in the conclusions (see line 355, 505-507).

    Furthermore, in order to address the doubt of the Reviewer concerning the propagation routine (also shared with the second Reviewer), we have included a completely new section (Section 5) to the revised manuscript. The new section specifically address the effect of masking on the performances of the algorithm in a systematic way. We are confident the reviewers will appreciate the new analysis.

10. One significant critique of this study is its reliance on a single case study: the Pakistan 2022 case study. This limited scope is insufficient, particularly considering the lack of easily accessible validation data for this specific case study. Moreover, it hinders the ability to compare the method's accuracy and computational efficiency with other existing approaches. To address this limitation, it is recommended to include additional case studies using published datasets. This would allow for a more convincing demonstration of the advantages of the proposed method.

    Unfortunately, freely available data sources of real-case scenarios are usually not openly available, especially when it comes to water depth. See for example the review paper mentioned in comment #3 (Teng et al. (2022)).

    Acknowledging the lack of validation data as a critical aspect, in order to address the comment of the Reviewer the revised manuscript includes a new section dedicated to assess FLEXTH against hydrodynamic simulations in 2 geographically different locations (Section 4). Although hydrodynamic simulations do not necessary reproduce real-world cases, they provide realistic physically-based scenarios useful for validation purposes and they readily provide flood extents and water depths, circumventing the limitation of remote-sensing methodologies. The section offered the possibility to further benchmark the performances of FLEXTH with those of FwDET, which can be considered as a reference in the field.

11. Lines 234-235, how does this run speed compare to other existing approaches? You are claiming computational advantages without solid evidence.

Thanks for pointing that out. Lines 340-345 on the revised manuscript now provide detailed information in this regard.

12. Line 243: Please clarify the meaning of CEMS. Please spell out acronyms before their first reference. Similarly, for EMSR629, FABDAM, and other acronyms, provide their full expansion before their initial mention.

Thanks for the remark. We clarified the meaning of CEMS (line 239). We don't deem necessary to extensively describe the other acronyms as references are provided.

13. You are using one satellite product to validate another satellite product of flood extent, which warrants clarification. Please explicitly state the advantage of CEMS over GFM flood extents for validation purpose.

This is true but it is the only possible way unless ground-based flood delineations are available (unrealistic) or if aerial-based flood delineation are provided (rare and with difficult data access). An alternative would be to use numerical models (which we did in the revised manuscript, see the response to comment #10). In the current text it is specified that the CEMS flood delineation are semiautomatic with expert supervision/refinement (see lines 241-243). On the other hand, as stated in the text (lines 52-55), GFM is a fully automatic and unsupervised system (with the consequent limitations). This, in our opinion, already stresses the higher accuracy of CEMS delineations.

14. In addition to the metrics listed in Table 2, it would be recommended to include F-stat as an additional accuracy metric.

Agreed, see the updated Table 2.

15. Using ICESat-2 altimetry data as truth to validate water depth estimates could be problematic due to mismatching of footprint and timing, as you have discussed in Section 4. To mitigate this concern, it would be advisable to incorporate additional case studies with more suitable validation data, as mentioned in the comments above.

The use of ICESat-2 to validate flood depth estimates is per se a novelty in the field which was probably not stressed sufficiently (as noted by the second Reviewer). In fact, we are not aware of any published study that uses similar methods. Errors due to the spatial mismatch are minimal (as discussed in the manuscript). Errors due to the temporal mismatch between GFM flood maps and ICESat-2 acquisitions are also mentioned and, as

correctly pointed out by the Reviewer. However, we would like to underline that similar problem would occur regardless the source of the satellite products used for validation. This happens, for example, when validating flood extent products with other products having higher mapping capabilities.

We are confident that the new Section 4 will help clarify these aspects. In fact, the section is dedicated to validate the methods by means of hydrodynamic simulations in 2 additional case studies.

16. Line 322 acknowledges the critical importance of DTM accuracy and resolution. However, it raises the question of why the study does not utilize high-resolution and high-accuracy DEM data, which are available in many regions globally. If data accessibility is an issue, it would be more beneficial to include additional case studies that utilize such data to enhance the robustness and applicability of the findings.

We believe that showing the performances of the methodology even without using site-specific high resolution topographical data is an additional proof for the suitability of the methods for large scale applications. As suggested for similar methods (see Cohen et al., Remote sensing, 2022) higher quality DTM will systematically improve the performances of the methodology.
That said, the new test cases that are included in the revised manuscript use DTM with different resolution (10 and 2 m), thus helping clarifying this aspect.

17. Thank you for providing the source code. After reviewing the code, I was unable to identify memory control or chunking algorithms that would support your claims of computational efficiency for large-scale studies. Could you please clarify this aspect?

Thanks for pointing that out. This aspect is now specified in line 227. We are considering adding in the FLEXTH repository a little script for tiling extremely large rasters. Note that performance metrics declared in in 340-345 of the revised version of the manuscript refer to the full rasters.

18. Validation data have not been provided along with the source code.

All the validation data are freely accessible online on GFM and ICESat-2 portals ( https://portal.gfm.eodc.eu/login?redirect=%5Bobject%20Object%5D ; https://openaltimetry.earthdatacloud.nasa.gov/data/index.html). FABDEM is also publicly accessible (https://data.bris.ac.uk/data/dataset/s5hqmjcdj8yo2ibzi9b4ew3sn). We don't see the point to include in a dedicated repository a duplicate of these bulky data (references

to the data sources are clearly stated in the manuscript). Nonetheless, what we are considering is to make available the result the hydrodynamic simulation performed for the first additional test case included in the new Section 4. We believe that the latter data is more critical given the notorious lack of validation/testing datasets for flood-related studies of this type.

**Reviewer 2**

The manuscript presents a new algorithm for improving and enhancing remote sensing-derived flood inundation maps. The algorithm was developed to address limitations in the Copernicus Global Flood Monitoring (GFM) system but can be used for other applications. The algorithm fills gaps in the flood maps by expanding the flooded domain into the GFM 'Exclusion mask' area and calculates water depth.  The approach builds nicely on recent efforts in this field and presents a considerable advancement. The manuscript is very well written and the algorithm evaluation analysis is well-reasoned and presented. I have two main concerns: (1) the evaluation benchmark is a flood map (and water depth) derived from different satellite sensors which likely suffers from somewhat similar (or other form of) biases as the SAR map; (2) the algorithm description is not as clear as it should be. The authors acknowledge the issues with the benchmark dataset used for evaluation and frame the results in this context. They also rightly assert that robust data for large-scale flooding is scarce. One remedy is to enhance the AOI-based analysis and add locations in which there is higher confidence in the benchmark data.

We thank the Reviewer for appreciating the study and to stress some limitations which will be addressed in the revised version. In particular, an additional section has been added specifically to address point (1) above (such aspect was also stressed by Reviewer 1). In this new section, the results from virtual flood scenari obtained via hydrodynamic simulations have been used as a reference to benchmark the estimates provided by FLEXTH. Hydrodynamic simulation in fact are suitable for the purpose because: i) provide a clear delineation of flooded areas (which is not always possible based on satellite products because of no sensitivity and/or water-look-alike conditions); ii) provide estimates of water depth and water level which can be used for benchmarking. We furthermore improved the description of the algorithm (point (2) above), particularly folloiwng the remarks provided by the Reviewer in the following comments and adding extensive references to the panels of Figure 2 throughout the main text.

Another new section has been added (Section 5) to the revised manuscript. The new section specifically focuses on assessing the performances of FLEXTH in case of progressively more invasive no-data regions. The section is added to address the interest/criticism displayed by both reviewers towards this aspect.

5 completely new figures support the new sections.

Specific issues:

Line 118: clarify 'last flooded pixel'

The sentence has been rephrased and clarified.

Line 127: what do you mean by '"morphological" closing'?

A reference for the technical term has been provided.

Section 2.2 is not clear enough.

We attempted to improve the section. However the section is quite long and complex and it is not specified what is unclear.

The small paragraph starting in line 132 is not very clear.

Thanks for pointing that out. We added a reference to a computer vision text which explain the terms used in the text (Gonzalez and Woods 2018).

Figure 2 can be better explained and referenced in the text.

Additional references are provided in the revised section 2.2 and 2.3. In particular, each panel of Figure 2 is now explicitly referenced and described in the revised manuscript.

Line 154: not sure what you mean by '4-connectivity'

The sentence has been removed because the technicality is not crucial and might just confuse.

Figure 3: clarify if these are purely synthetic (1D) results (i.e. not using the full algorithm)

Agreed, the aspect should be clear now from the caption of the figure.

I don't recall you explicitly explaining when the recursion ends (I assume when water cannot be further propagated). It is also not entirely clear the 'seeding' of the reduction - is every pixel at the edge of the flood a seed?

The revised text explicitly states these aspects (see lines 186-195).

Table 2 and text: consider reporting the overall improvement in terms of %.

Thanks for the good suggestion. The % change has been included in Table 2.

Line 235: how many pixels and what is the resolution used?

The information is now provided (see lines 218-219)

Line 272: I don't understand what 'i=1,2' stands for.

They denote the two tracks of points acquired by ICESat-2 (line 267).

Line 289: consider removing 'To conclude'

Agreed.

Figure 6: what is RMSE represent - error compared to what?

Thanks for pointing that out. The comparison is with respect to the ideal case (Delta = 0). This is now highlighted in the revised caption of the figure.

The use of ICESat-2 altimetry data to evaluate the water depth prediction is quite novel (to my knowledge) and could be of great interest. The limitations in the data acquisition and processing is well described but additional emphasis needs to be made on the limitations and unknowns associated with your approach - highlighting the need for additional research focused on this approach.

Thanks for appreciating the novelty of the methodology. We addressed the comment by providing some additional considerations in the revised Discussion (lines 388-400)

Line 394: 'hydraulic connectivity' is a good term to use in this context - consider adding it to the algorithm description.

We agree, thanks for pointing that out!

Line 403: 'FLEXTH can [also] be effectively applied to...'

Agreed.

The algorithm applies the expansion procedure to masked areas which you found to be a potential limitation. Did you look into running it without this limit?

The effect of the exclusion mask on the performances of FLEXTH and particularly on its propagation routine has been systematically assess the new dedicated Section 5. We deem this as an interesting aspect which was also stressed by the first reviewer. We are confident that the reviewers will appreciate the new analysis!

---

## Author Response (AR2)

**Response to S. Cohen (Reviewer # 1)**

**Manuscript title: Water depth estimate and flood extent enhancement for satellite-based inundation maps**

The authors thoroughly addressed the reviewers' comments. The two new sections add great value to the manuscript.

We thank the reviewer for appreciating our efforts in improving the study.

Minor comments:
1. The new sections need to be better integrated into the manuscript by moving the methodology and results description to the paper's methodology and results section. They were added as a somewhat stand-alone segment.

We see the point of the reviewer (we also noticed this aspect and we have been thinking about it). However we would like to dedicate the method section exclusively for the description of the FLEXTH methodology (which is the focus of the study). Furthermore, if we were to address the concern raised in the review, it would be necessary to include the ICESat-based flood depth evaluation procedure in the methods section as well. In our opinion this would reduce the readability of the text.

2. In section 4: 'bypassing [some] of the limitations...' Stress that simulated floods have their own limitations and (often considerable) biases.

Agreed.

3. FwDET version 2.1 (Cohen et al., 2022) was also implemented in GEE, providing better boundary cell filtration. Even if the authors do not use the most recent version it is useful to reference it for the readers.
We have to admit it is not super clear to navigate through all the versions of FwDET, which also include alternative implementation for different platforms (e.g. Arcgis, GEE, Qgis).

Nonetheless, the GEE version we retrieved from Github (the one we used for benchmarking) features both the recursive smoothing filtering as well as the slope filter of the boundary cells (see lines 335-340). Thus, suggesting it is in fact the latest implementation of the algorithm (i.e. the Cohen et al., 2022 mentioned by the reviewer). This aspect is now clarified in the manuscript and further citations are provided (see lines 299-300).

Cohen, S., B.G. Peter, A. Haag, D. Munasinghe, N. Moragoda, A. Narayanan, S. May, (2022). Sensitivity of Remote Sensing Floodwater Depth Calculation to Boundary Filtering and Digital Elevation Model Selections. Remote Sensing, 14, 5313. https://doi.org/10.3390/rs14215313